# When Perplexity Lies: Generation-Focused Distillation of Hybrid Sequence Models

## Abstract

Converting a pretrained Transformer into a more efficient hybrid model through distillation offers a promising approach to reducing inference costs. However, achieving high-quality generation in distilled models requires careful joint design of both the student architecture and the distillation process. Many prior distillation works evaluate downstream multiple-choice benchmarks by ranking candidate answers with log-likelihood rather than requiring autoregressive generation, which can obscure important differences in model quality. For example, on overlapping benchmarks, we show that a 7B parameter distilled model that nearly matches its teacher to within 0.2 pp under log-likelihood scoring actually falls behind by 20.8 pp when the model must generate answers autoregressively.

We investigate this phenomenon with GenDistill, a multi-stage pipeline we designed for distilling a pretrained Transformer into an efficient Hybrid Kimi Delta Attention (Hybrid-KDA) student. Using it as a controlled testbed on Qwen3-0.6B, we systematically ablate six design axes (training objective, loss masking, training duration, dataset selection, parameter freezing, and architecture choice) and evaluate every choice under both log-likelihood and generation-based protocols. We find that log-likelihood-based evaluation consistently underestimates the gap between teacher and student, and can in some cases reverse the ranking of design choices, so conclusions drawn from perplexity-only evaluation may be misleading. Among the factors we study, dataset selection, completion-only masking, and freezing attention layers during post-training have the largest impact on generation quality.

Our best distillation recipe, using a Hybrid-KDA model as the student, retains 86–90% of teacher accuracy on knowledge benchmarks while reducing KV cache memory by up to 75% and improving time-to-first-token by 2–4× at 128K-token contexts.

## 1 Introduction

Transformer-based language models dominate modern NLP due to their strong accuracy and scaling properties. However, self-attention incurs $\mathcal{O}(n^2)$ time and memory complexity in the sequence length $n$, and practical deployments additionally pay a linearly growing key-value (KV) cache cost during autoregressive inference.

In parallel, efficient sequence models with linear-time recurrent-style computation, including state space models (SSMs), have emerged as an attractive alternative. These architectures can offer linear-time processing and $\mathcal{O}(1)$ *per-step* memory growth during generation (i.e., no KV cache), enabling faster decoding and a smaller memory footprint in long-context settings (Gu & Dao, 2023; Yang et al., 2023; 2025c; Dao & Gu, 2024).

Because training large transformers from scratch is computationally expensive (often on the order of trillions of tokens), a growing line of work studies *cross-architecture distillation*: transferring capabilities from a pretrained transformer teacher into a more efficient recurrent student (Wang et al., 2024; Bick et al., 2025a; 2024; Goldstein et al., 2025).

Despite encouraging results, prior methods for distilling Transformers into SSMs both optimize for and evaluate on perplexity, without targeting generation quality at any stage of the pipeline. While the broader NLP community has long recognized that perplexity is an imperfect proxy for generation quality, this gap takes on particular severity in cross-architecture distillation. The crux of the issue lies in how models are evaluated. *Perplexity-based* evaluation computes the log-likelihood of each candidate answer conditioned on the prompt and selects the highest-scoring option, testing whether the model *assigns high probability* to correct answers. *Generation-based* evaluation requires the model to produce answers autoregressively, after which outputs are typically parsed from an expected format or verified with task-specific criteria (e.g., exact-match answers or executable code), testing whether the model can *generate* correct answers. In this work, we use LM Evaluation Harness (Gao et al., 2024) for perplexity-based evaluation and EvalScope (ModelScope Team, 2024) for generation-based evaluation. The distinction is consequential because a model can rank correct answers highly without being able to generate them. To illustrate the severity of this discrepancy, we evaluate QRWKV6-7B-Instruct (Lin et al., 2025b), distilled from Qwen2.5-7B-Instruct, under both evaluation protocols. Figure 1 contrasts the perplexity-based results reported by the original paper with our generation-based evaluation of the same model checkpoint (full numbers in Table 5, Appendix A).

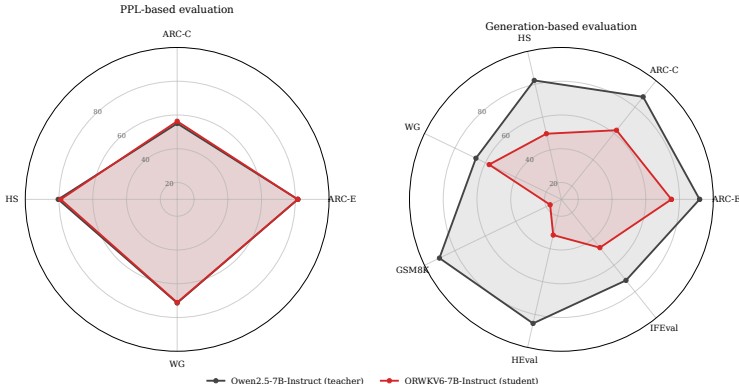

Figure 1: Evaluation protocol gap for QRWKV6-7B-Instruct (distilled from Qwen2.5-7B-Instruct), with both panels evaluating the same checkpoint. Perplexity-based numbers (left) are taken from the original paper (ARC-E, ARC-C, HS, WG); generation-based numbers (right) are our own and additionally include GSM8K, HEval, and IFEval. Full values appear in Table 5 (Appendix A).

The gap between the two protocols is substantial. As shown in Figure 1, QRWKV6-7B-Instruct nearly matches its teacher under PPL-based evaluation (average gap of just 0.2 pp), yet suffers catastrophic drops in generation-based performance (average gap of 20.8 pp). The pattern is representative of the broader literature, where cross-architecture distillation pipelines typically optimize on pretraining data and report their main downstream results via log-likelihood-based evaluation (e.g., LM Evaluation Harness) (Wang et al., 2024; Bick et al., 2025a; 2024; Goldstein et al., 2025). This gap motivates a systematic investigation of what it takes to produce distilled models with genuine generation quality. While works such as M1 (Wang et al., 2025), and recently MiniCPM-SALA (Team et al., 2026) have aimed to improve the reasoning ability of distilled linearized models, they rely on additional SFT and RL stages to improve the performance of the student model. These stages themselves can be quite significant (e.g. 1T tokens in the MiniCPM-SALA).

In contrast, we investigate the fundamental distillation choices themselves and how to maximize the generation quality of the distilled model without additional continual training. In this work, we develop GenDistill, a multi-stage distillation pipeline, making each of its design choices by empirically evaluating how that choice affects the student's downstream instruction-following and reasoning performance. We then obtain the strongest student by pairing the pipeline with a Hybrid Kimi Delta Attention (Hybrid-KDA) architecture.

Our main contributions are:

- We show that perplexity-based evaluation is a systematically misleading proxy for generation quality in cross-architecture distillation. The divergence appears consistently across every design axis we vary, and holds at both 0.6B and 1.7B parameters.

- To establish this systematically, we develop GenDistill, a multi-stage distillation pipeline whose design space we vary one axis at a time over a fixed teacher (training objective, loss masking, training duration, dataset selection, parameter freezing, and architecture choice). We will release our model code and distilled checkpoints upon publication.

- Based on these empirical results, we propose a concrete distillation recipe that maximizes the generation quality of the student model, combining knowledge distillation with completion-only masking, teacher-aligned instruction data, and frozen attention layers, which departs from the recipe a perplexity-guided search would have selected.

The remainder of the paper is organized as follows. Sections 3.1 and 3.2 describe the Hybrid-KDA architecture and the GenDistill distillation stages. Section 4 defines the two evaluation protocols whose disagreement is our subject. Section 5 then walks through each design choice and reports it under both protocols, where the recurring outcome is that perplexity and generation disagree, often reversing which choice looks better. The recipe we recommend is what this design space yields once the choices are made under generation rather than perplexity.

## 2 Related Work

The computational demands of Transformer-based language models have motivated substantial research on converting pre-trained models to architectures with subquadratic complexity. Chronologically, this line of work first focused on fine-tuning-based approaches that directly modify attention mechanisms and continue training with standard language modeling objectives, and later expanded to distillation-based approaches that transfer knowledge from Transformer teachers to structurally different student models.

Early works focused on replacing the softmax attention layer with expressive linear kernels, followed by fine-tuning the resulting model for 1–40B tokens (Chen et al., 2024; Kostelec & Guo, 2025; Zhang et al., 2024; Mercat et al., 2024). Notably, LoLCats (Zhang et al., 2025) achieved conversion with as few as 40M tokens via a two-stage approach: 20M tokens for attention alignment followed by 20M tokens to train LoRA adapters to recover model performance. However, these methods still exhibited a performance gap, particularly on benchmarks such as MMLU.

More recent works have leveraged multi-stage distillation for model conversion and generally report substantially stronger student performance. Table 1 summarizes the pipelines most related to ours. MOHAWK (Bick et al., 2024) introduced the bottom-up three-stage template of attention-matrix orientation, hidden-state alignment, and end-to-end knowledge distillation. Similar pipelines were proposed by Wang et al. (2024) (focusing on hybrid models and adding SFT and DPO post-training stages), Bick et al. (2025a) (scaling MO-HAWK to larger models), and Yang et al. (2025b) (distilling into a hybrid model with multi-head latent attention and Mamba layers). A concurrent work (Goldstein et al., 2025) proposed a highly efficient distillation process, using only 700M tokens while reporting state-of-the-art results on downstream tasks, however, they focused on purely recurrent models and, like the other pipelines, did not contrast likelihood-based with generation-based evaluation.

One fundamental limitation of SSMs and linear-attention models is their fixed hidden-state size, which can significantly degrade performance on in-context retrieval tasks. Consequently, many works have explored hybrid models to mitigate this gap. Two directions are relevant here: intra-layer hybrids, which replace only a subset of attention heads within a layer with SSM heads, and inter-layer hybrids, which replace entire attention layers with SSM layers. Although intra-layer hybrids allow a finer-grained choice of which attention heads to keep and which to convert to SSMs (Zuo et al., 2025; Dong et al., 2024), they can increase system complexity and inference overhead, complicating distributed parallelism optimizations (Team et al., 2025). For inter-layer hybrids, the choice of which attention layers are kept intact is crucial, as some layers appear to be critical for retrieval performance (Bick et al., 2025b).

Table 1: Multi-stage cross-architecture distillation pipelines. GenDistill follows the same bottom-up align-then-distill structure as this line of work; what is specific to it is the selection of every recipe choice under generation-based evaluation, a hybrid Kimi Delta Attention student, and a completion-only-masked instruction-distillation stage on teacher-aligned data. Token counts are as reported by each work under its own teacher and data.

| Pipeline | Student | Hybrid | Stages | Tokens | Instruction stage | Headline eval |
|---|---|---|---|---|---|---|
| MOHAWK | Mamba2 | No | matrix → hidden → KD | 3–5B | none | likelihood |
| Mamba-in-Llama | Mamba2 | Yes | transfer → KD → SFT → DPO | 20B | SFT + DPO | likelihood |
| Llamba | Mamba2 | No | matrix → hidden → KD | 8–12B | KD | likelihood |
| RADLADS | RWKV-6/7 | No | transfer → hidden → KD → SFT | 700M | none | likelihood |
| **GenDistill (ours)** | **KDA** | **Yes** | align → hidden → KD → KD | 1B | **completion-only KD** | **generation** |

Prior approaches have selected attention layers uniformly (Lieber et al., 2024; De et al., 2024; Bick et al., 2024). SMART (Yang et al., 2025b) studied the sensitivity of each layer (measured by the reduction in teacher–student KL divergence) when swapping a linear layer with an attention layer, and then heuristically selected layers to maximize total sensitivity. PostNAS (Gu et al., 2025) used a complex search procedure, training a once-for-all supernet and using beam search to find the optimal $K$ layers for specific downstream tasks. A concurrent approach proposed greedily adding attention layers back to a distilled fully linear model by measuring the KL divergence of the output logits between the resulting hybrid model and the original teacher model after distilling the teacher into the hybrid model (Li et al., 2025).

## 3 The GenDistill pipeline

Producing a high-quality distilled model requires jointly designing the student architecture and the distillation process. This section describes both components: the Hybrid-KDA student architecture (Section 3.1), including a beam-search method for selecting which layers retain softmax attention (Section 3.1.1), and the multi-stage distillation process that progressively aligns the student with the teacher (Section 3.2).

### 3.1 Student Model Architecture

To preserve the foundational knowledge acquired during pretraining, our student model inherits the majority of its architecture directly from the pretrained teacher model. Specifically, the vocabulary, embedding layers, layer normalizations, and multi-layer perceptrons (MLPs) are initialized from the teacher, while select attention layers are replaced with more efficient sequence mixers to improve inference efficiency.

The choice of sequence mixers used in the student model is crucial for the success of the distillation pipeline, both in terms of final student model performance, as well as how efficiently we can distill the transformer into the new architecture. Recent work has proposed several linear sequence mixers that maintain efficiency while improving model expressivity (e.g. Gated DeltaNet (Yang et al., 2025c), Kimi (Team et al., 2025), RWKV7 (Peng et al., 2025)). In this work, we focus on two linear architectures: Kimi Delta Attention (KDA) as our primary student mixer, and Mamba2 (Dao & Gu, 2024) as a baseline for comparison. We select Mamba2 because it is one of the most widely adopted SSMs and has been used as the student architecture in several prior distillation works (Phi-Mamba (Wang et al., 2024), Llamba (Bick et al., 2025a)). We select KDA because of its strong reported performance. KDA refines Gated DeltaNet with a per-channel rather than scalar forget gate, and we give the full recurrence in Appendix B.

Additionally, there are several smaller architectural choices that have also proven important for a good and efficient distillation process. We enumerate them here, while an ablation is available in Appendix F. Similar to Bick et al. (2024), we observe that removing the output norm in the SSM improves the distillation process. We also use identity initialization of the short convolutions and the output gating in the Kimi layer, however, contrary to the findings in Team et al. (2025), we find that using SiLU gating is better than using Sigmoid

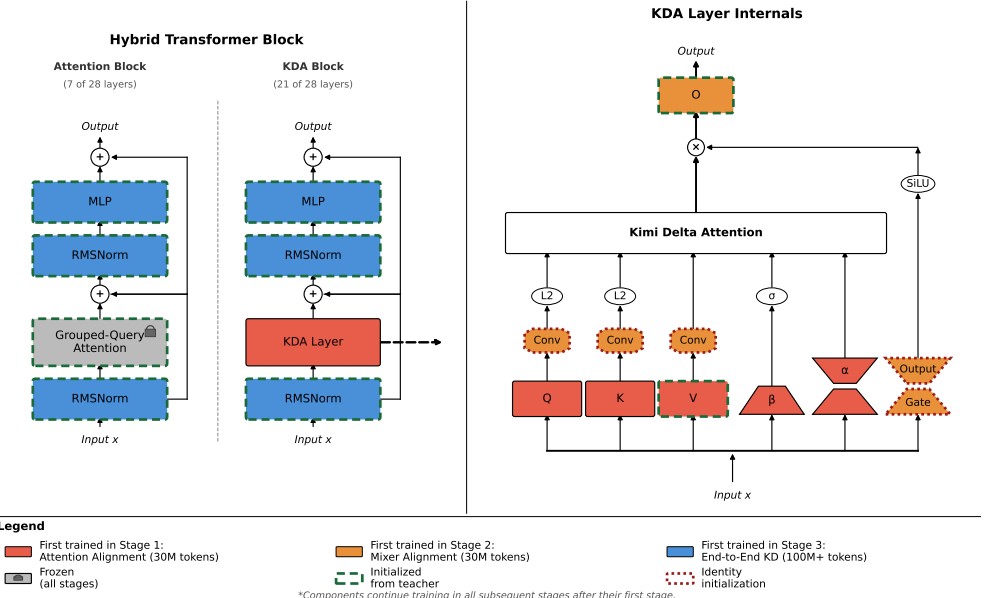

Figure 2: Architecture overview. **Left:** The two block types in the hybrid transformer: an Attention block (7 of 28 layers, frozen from the teacher) and a KDA block (21 of 28 layers). MLPs and layer norms are initialized from the teacher and trained only during end-to-end distillation (Stage 3). **Right:** Internal structure of the KDA layer. Fill colors indicate the earliest distillation stage in which each component begins training; once introduced, components continue training in all subsequent stages. Green dashed borders denote initialization from teacher weights; dark-red dotted borders denote identity initialization; the absence of a custom border denotes random initialization.

gates in the distillation setting. Since we are distilling from the Qwen3 model family (Yang et al., 2025a), which uses QK norms in the attention layer, we also keep normalization of QK values in the Kimi layer[1].

### 3.1.1 Attention layer selection

In hybrid architectures, the selection of which layers retain softmax attention versus those replaced with linear alternatives significantly impacts model performance. While certain attention heads appear universally important for in-context retrieval (Bick et al., 2025b), our initial experiments revealed that optimal layer selection is more task-dependent.

Greedy layer selection may yield suboptimal configurations, as it fails to account for interactions between layers. To address this limitation, we employ beam search to explore layer combinations more thoroughly, similar to the approach in PostNAS (Gu et al., 2025). However, unlike PostNAS, we avoid training an extensive supernet by leveraging only the pretrained teacher model and a fully linear student model (comprising Kimi layers) obtained through our distillation pipeline.

Our search procedure, *Beam Search (Add)*, initializes from the fully linear student and iteratively restores attention layers from the teacher, retaining at each step the configurations that achieve the lowest perplexity. To account for task dependence, we evaluate perplexity on three complementary datasets: (i) a synthetic associative recall (AR) task that probes in-context retrieval, (ii) C-EVAL for Chinese language understanding, and (iii) FineWeb-EDU for general language modeling, and select layers based on the average across all three.

Training a student per candidate would be the ideal ranking signal but is prohibitively expensive, so we instead rank by perplexity as a cheap comparative proxy. This within-pipeline use scores an undertrained scaffold to decide only which layer positions matter, and is distinct from the final-evaluation use of perplexity

---

[1]We use L2 normalization without learnable scales, following the Kimi Delta Attention layer implementation in Yang & Zhang (2024).

that this paper argues against. Appendix E expands on this distinction, positions our proxy against the compute-heavy PostNAS and GA-S2 (Li et al., 2025) alternatives, reports the resulting layer selections, and shows that the cheap signal does not cost selection quality on Qwen3-0.6B.

## 3.2 Distillation Stages

GenDistill uses a multi-stage distillation process designed to progressively align the hybrid student model with the Transformer teacher. Direct end-to-end distillation can be inefficient under architectural mismatch, so we follow the bottom-up multi-stage blueprint of MOHAWK (Bick et al., 2024), with Stage 1 adapted to KDA by aligning hidden states rather than attention logits (KDA cannot efficiently materialize the full $N \times N$ attention matrix). We then perform end-to-end KD in Stage 3, separating pretraining-data distillation (Stage 3a) from instruction-tuning distillation (Stage 3b). We report generation-based evaluation alongside perplexity to capture quality differences that are not visible under likelihood-only protocols. Overall, using KDA as the student mixer allows us to operate at a smaller total token budget ($\sim$1B) than prior multi-stage distillation pipelines (MOHAWK: $\sim$3B; Llamba: $\sim$8B (Bick et al., 2025a)).

### 3.2.1 Stage 1: Initialization and Attention Matrix Alignment

The student model inherits the embedding, normalization, and MLP layers directly from the teacher. For the sequence mixers, we replace specific attention layers with Kimi Delta Attention (KDA) layers. We initialize the Value ($V$) and Output ($O$) projections of the KDA layers using the corresponding weights from the teacher, exploiting the architectural similarity, see Figure 2. However, we find that initializing the Query ($Q$) and Key ($K$) projections from the teacher does not necessarily improve the training, and are thus initialized randomly.

While MOHAWK (Bick et al., 2024) aligns the full attention logits to approximate the teacher's attention patterns, KDA cannot efficiently materialize the full $N \times N$ attention matrix. Instead, we minimize the L2 distance between the hidden states of the attention layers of the teacher and student. Only the $Q$ and $K$ projections are trained. We use a filtered subset of FineWeb-EDU and FineWeb-Edu-Chinese-2.1 (score $\geq 4$)[2], training for 30M tokens with a sequence length of 2048. We employ a warmup-stable-decay learning rate schedule (max LR $5e - 4$, 10% warmup). Samples are packed into fixed sequence lengths with cross-document masking (Zuo et al., 2025).

### 3.2.2 Stage 2: Block-wise Hidden State Alignment

The second stage aligns the feature representations of the student and teacher blocks independently. We minimize the L2 distance between the outputs of the student's mixer block and the teacher's attention block. In this stage, all weights within the sequence mixer are trainable, while LayerNorms remain frozen as they are initialized from the teacher. We train for another 30M tokens using the same dataset and schedule as Stage 1, but with a reduced learning rate ($1e - 4$).

Crucially, we find that this two-step alignment (Stage 1 followed by Stage 2) is superior to training the mixer all at once or initializing all weights from the teacher. Skipping Stage 1 results in significant perturbations to the initialized $V$ and $O$ weights, destabilizing the training. By the end of this stage, after only 60M tokens and without any end-to-end training, the student model achieves a validation perplexity of $\sim$25, approaching the teacher's $\sim$15–20. Other initialization stages either reach much higher perplexity (MOHAWK (Bick et al., 2024): $\sim$1000 after 240M tokens) or run far longer (Llamba (Bick et al., 2025a): 3B tokens).[3]

### 3.2.3 Stage 3: End-to-End Knowledge Distillation

Block-wise alignment treats layers independently, leading to error accumulation when layers are composed during inference. To address this and recover coherent text generation capabilities, we perform end-to-

---

[2]Datasets: Fineweb-EDU (score $\geq 4$) and Fineweb-Edu-Chinese-2.1 (filtered subset).

[3]These perplexities use different teachers and evaluation datasets and are not directly comparable to ours, though all are measured on large web-scale pretraining corpora.

end knowledge distillation (KD). We minimize the forward KL divergence between the teacher and student output logits. We freeze the attention layers and input embeddings, training all other parameters.

This stage is divided into two phases. First, we distill on the pretraining dataset to couple the layers and smooth internal representations (Stage 3a; we select the token budget in Section 5.2). Second, to adapt the model for instruction following, we switch to the AM-Qwen3-Distilled dataset (Tian et al., 2025)[4] (Stage 3b). We increase the sequence length to 4096 and use a learning rate of $2.5e - 5$. The Stage 3b design choices, in particular the training objective and the loss-masking scope, are exactly where perplexity and generation-based evaluation diverge, so we select each one under generation and report it under both protocols in Section 5.1. We separate these two phases so that the student first recovers general language modeling over a broad distribution before acquiring instruction-following and reasoning from the teacher.

## 4 Evaluation Setup

Having described the Hybrid-KDA architecture and GenDistill pipeline (Section 3), we now turn to evaluation. A central design choice in our evaluation is the use of *two complementary protocols* that, as we show in Section 5.1, can tell contradictory stories about model quality:

- **Generation-based evaluation**: The model generates answers autoregressively. Responses are scored against reference answers. This protocol tests whether the model can *produce* correct answers, i.e. the practical use case for deployed models.

- **Log-likelihood ranking on multiple-choice tasks**: The model scores each candidate answer by its log-likelihood conditioned on the prompt, and the highest-scoring candidate is selected. This protocol tests whether the model assigns high probability to correct answers, which can diverge substantially from generation quality.

In our implementation, we use EvalScope (ModelScope Team, 2024) for generation-based evaluation and LM Evaluation Harness (Gao et al., 2024) primarily for the log-likelihood-based multiple-choice evaluations.

**Benchmarks.** We evaluate on short-context benchmarks across five categories: **language modeling** (LAMBADA (Paperno et al., 2016)); **common sense** (HellaSwag (Zellers et al., 2019), WinoGrande (Sakaguchi et al., 2021), PIQA (Bisk et al., 2020)); **knowledge** (MMLU-Redux[5] (Gema et al., 2024), C-Eval (Huang et al., 2023), CMMLU (Li et al., 2023)); **reasoning** (ARC (Clark et al., 2018), BBH (Suzgun et al., 2023), GSM8k (Cobbe et al., 2021), HumanEval (Chen et al., 2021)); and **instruction following** (IFEval (Zhou et al., 2023)). We evaluate tasks in both frameworks where available to analyze evaluation discrepancies[6]. We additionally evaluate long-context quality using LongBench (Bai et al., 2023) and the NIAH-single subtasks from RULER (Hsieh et al., 2024)

## 5 Results

We organize the results around the central finding. We first show that log-likelihood-based and generation-based evaluation disagree on the design choices that strongly affect the student, the training objective, the loss-masking scope, and attention freezing, and that this disagreement persists at 1.7B parameters (Section 5.1). Treating generation as the reliable criterion, we then fix the remaining recipe choices, the post-training dataset, the pretraining-distillation budget, and the training duration (Section 5.2). Finally, we compare the architectures this recipe produces, confirm that the selected architecture also scales to 1.7B, and quantify their efficiency (Section 5.3).

---

[4]https://huggingface.co/datasets/a-m-team/AM-Qwen3-Distilled

[5]We use MMLU-Redux (Gema et al., 2024) for generation-based evaluation (EvalScope), as it corrects a number of erroneous questions identified in the original MMLU. For perplexity-based evaluation (LM Evaluation Harness), we use the standard MMLU, since MMLU-Redux is only available in a generative format in that framework. The two benchmarks are tightly correlated in terms of performance.

[6]For BBH, HumanEval, GSM8K the performance on the lm-evaluation-harness was trivially low, around 0, which is likely an issue with the task setup, so we do not report these tasks in the lm-evaluation harness framework.

## 5.1 Where perplexity and generation disagree

Three design choices strongly shape the distilled student. They are the training objective, the loss-masking scope, and whether the retained attention layers are frozen during instruction tuning. We evaluate each under both protocols, log-likelihood-based (LM Evaluation Harness) and generation-based (EvalScope). On all three, the two protocols disagree, and in places reverse the ranking of the same pair of models, with perplexity consistently understating differences that generation makes plain. We select each choice under generation.

Figure 3 profiles the four (objective, masking) configurations and the teacher under both protocols. Under perplexity the four students sit almost on top of one another and on top of the teacher. Under generation they spread apart and fall well below the teacher, and they separate further on the reasoning, code, and instruction-following tasks that perplexity cannot score at all (full per-benchmark numbers in Table 17, Appendix K).

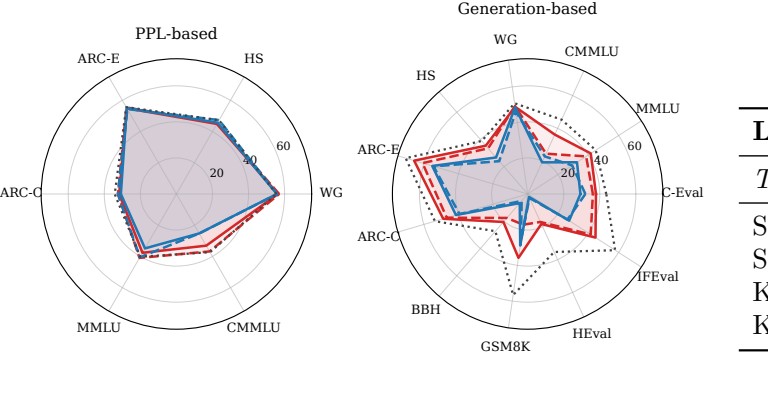

| Loss | Mask | PPL | Gen |
|------|------|-----|-----|
| *Teacher* | | 45.0 | 50.7 |
| SFT | P+C | 42.6 (−2.4) | 36.0 (−14.7) |
| SFT | C | 41.2 (−3.8) | 37.3 (−13.4) |
| KD | P+C | 44.6 (−0.4) | 42.2 (−8.5) |
| KD | C | 43.1 (−1.9) | 46.1 (−4.6) |

Figure 3: Stage 3b ablation at the 250M-token budget: the four student configurations (SFT and KD, each with full-sequence P+C or completion-only C masking) and the Qwen3-0.6B teacher, under both evaluation protocols (accuracy in %). **Left:** spider profiles. The perplexity panel covers only the six benchmarks that log-likelihood scoring can produce, where the four students sit almost on top of one another and on the teacher, while the generation panel adds the reasoning, code, and instruction-following tasks that have no perplexity counterpart and spreads the configurations apart. **Right:** average accuracy over the six benchmarks common to both protocols (WG, HS, ARC-E, ARC-C, MMLU, CMMLU), with the gap to the teacher shown in red. Perplexity places every student within 3.8pp of the teacher, whereas generation opens gaps of 4.6 to 14.7pp and re-ranks them: SFT looks competitive under perplexity but drops furthest under generation. Full per-benchmark numbers, including ranking reversals on HellaSwag, ARC-C, and MMLU, are in Table 17, Appendix K.

**KD vs. SFT.** The training objective strongly affects generation quality: KD is consistently performing better than SFT across generation benchmarks (Table 17), with its advantages most visible on reasoning, code generation, and instruction following. By contrast, SFT often appears competitive under perplexity-based evaluation, highlighting that next-token likelihood on post-training data is not a reliable proxy for answer generation quality. On HellaSwag the disagreement is an outright inversion, with SFT attaining lower perplexity than KD but markedly lower generation accuracy. Appendix N characterises the qualitative shape of this disagreement on the completion-only 500M-token pair, where the SFT generations exhibit multiple pathologies that are essentially absent from KD: template wrapping, multiple-choice position collapse, wordier-but-less-accurate math chains, and recency mimicry of in-context demonstrations.

**Loss masking.** Which tokens contribute to the training loss matters as much as the objective itself. When the loss is computed over the full sequence, SFT can match or exceed KD on some benchmarks, especially

those that emphasize language modeling (e.g. Lambada). By contrast, computing the loss only on completion tokens consistently improves generation quality for both objectives (Table 17), with KD benefiting more from this choice. The trade-off is that language modeling ability improves more slowly, since the loss is applied to fewer tokens.

Table 2: Effect of freezing attention layers during Stage 3b. Generation-based evaluation (left) reveals large improvements from freezing ($\overline{\Delta}$=5.2pp), while perplexity-based evaluation (right) shows negligible differences ($\overline{\Delta}$=0.5pp). Best result per column in **bold**.

| | Generation-based (EvalScope) | | | | | | Perplexity-based (LM-Eval) | | | | | |
|---|---|---|---|---|---|---|---|---|---|---|---|---|
| **Frozen?** | WG | HS | ARC-E | ARC-C | C-Eval | MMLU-R | WG | HS | ARC-E | ARC-C | CMMLU | MMLU |
| No | 40.7 | 17.5 | 50.0 | 37.7 | 35.7 | 35.8 | 50.6 | 39.9 | 42.2 | 29.4 | 35.9 | **29.7** |
| Yes | **47.5** | **23.7** | **57.8** | **43.5** | **37.3** | **39.0** | **51.2** | **40.0** | **43.4** | **30.0** | **36.7** | 29.5 |

**Freezing attention.** A natural question is how to preserve the quality of the retained attention layers during instruction tuning. Table 2 compares frozen versus unfrozen attention layers during Stage 3b under both evaluation protocols. Freezing attention weights significantly improves generation-based performance across every benchmark. Unfrozen attention layers allow the instruction-tuning loss to overwrite pretrained representations, causing catastrophic forgetting of the reasoning capabilities acquired during pretraining. Critically, perplexity-based evaluation almost entirely masks this effect. The same frozen/unfrozen comparison that produces 5–8pp generation improvements yields less than 1pp differences on perplexity metrics ($\Delta$-row, Table 2, right). We further investigate whether MLP parameters exhibit the same sensitivity in Appendix I.

**Persistence at 1.7B.** We verify the three ablations at the 1.7B scale, distilling from a Qwen3-1.7B teacher into the same Hybrid-KDA architecture. The protocol disagreement reproduces: KD again leads SFT on generation while SFT attains the lower perplexity, and freezing again improves generation while perplexity is nearly unchanged. Completion-only masking is the one axis that does not transfer cleanly. It keeps its advantage on instruction following, code, and reasoning, but loses the commonsense multiple-choice benchmarks to full-sequence masking, where the completion-only student produces near-chance answers consistent with an extraction failure rather than a loss of ability. Full results are in Appendix M.

## 5.2 Distillation recipe

Section 5.1 fixes three choices, the training objective (KD), the loss-masking scope (completion-only), and freezing the retained attention layers during Stage 3b. Two choices remain, the post-training dataset and the token budget of each distillation stage, and we select them under generation as well.

For the dataset choice, we compare six instruction corpora, AM-Qwen3-Distilled (Tian et al., 2025), SFTv3 (Wang et al., 2024), Smoltalk2 (Bakouch et al., 2025), OpenHermes 2.5 (Teknium, 2023), Nemotron Post Training Dataset v1 (Nathawani et al., 2025), and Chinese-Instruct (Zhang, 2025), each under the same configuration (Hybrid-KDA student, KD, 500M-token budget, varying only the Stage 3b data source). The dataset turns out to be the largest single lever in the recipe, with average generation accuracy ranging from 26.2% to 46.9% and individual tasks moving by as much as 30.5 pp on ARC-E (Table 3). AM-Qwen3-Distilled, which contains Qwen3-generated responses verified for correctness, performs best by a wide margin, consistent with prior evidence that distilling on data close to the teacher's distribution transfers more effectively (Goldstein et al., 2025; Tian et al., 2025). We adopt it for all subsequent runs.

We also study how many tokens to spend on each distillation stage. Adding Stage 3a tokens keeps reducing held-out KL, but these loss improvements largely do not lead to better downstream generation after Stage 3b (Appendix J). Training Stage 3b longer leaves average generation roughly flat, as gains on Chinese, instruction-following, and code benchmarks offset declines on English ones (Appendix K). We use 500M tokens for each stage, capturing the modest Stage 3a gains on the tasks that do respond while balancing the Stage 3b trade-offs.

Table 3: Effect of the Stage 3b dataset on downstream performance (generation-based evaluation). All runs use the same token budget (500M); Avg is the mean over the six benchmarks.

| Distillation dataset | C-Eval [%] | ARC-E [%] | ARC-C [%] | HS [%] | WG [%] | MMLU-redux [%] | Avg [%] |
|---|---|---|---|---|---|---|---|
| SFTv3 | 39.1 | 64.3 | 47.8 | 24.8 | 40.4 | 40.5 | 42.8 |
| Chinese-Instruct | 18.0 | 34.9 | 25.8 | 16.4 | 37.8 | 24.3 | 26.2 |
| AM-Qwen3-Distilled | **39.8** | **65.4** | **49.3** | **37.2** | 48.9 | **40.8** | **46.9** |
| Smoltalk2 | 37.9 | 63.8 | 46.3 | 33.2 | **50.0** | 37.9 | 44.9 |
| Open Hermes | 37.0 | 49.1 | 33.9 | 13.4 | 42.0 | 35.8 | 35.2 |
| Nemotron Post Training v1 | 33.1 | 57.0 | 41.9 | 24.9 | 42.8 | 37.0 | 39.5 |

## 5.3 Architecture comparison

We now compare the architectures produced by the full GenDistill pipeline, evaluating how the choice of sequence mixer and the retention of attention layers (Hybrid vs. Pure) affect downstream generation quality. We distil five sequence mixers under the identical pipeline, spanning a range of the linear-attention design space: KDA, Mamba2, Gated DeltaNet (Yang et al., 2025c), GLA (Yang et al., 2023), and Lightning Attention (Qin et al., 2024). Mamba2 serves as the established baseline, being one of the most widely adopted SSMs and used in prior distillation work (Phi-Mamba, Llamba). To isolate architectural effects, we change only the sequence mixer and whether attention layers are retained, keeping all other hyperparameters fixed. The design choices behind the pipeline (pretraining budget, dataset, objective, and parameter freezing) are established in the preceding Sections 5.1 and 5.2, and full training and configuration details are provided in Appendix D.

Table 4: Comparison of distilled architectures (generation-based evaluation, EvalScope). "Hybrid" models retain (the selected) 7 attention layers, "Pure" models use 0. The lower block repeats the selected Hybrid-KDA architecture and recipe at 1.7B against a Qwen3-1.7B teacher. Results are averaged over evaluation seeds (3 at 0.6B, 2 at 1.7B). Best 0.6B student result per column in **bold**; the 1.7B rows are compared to their own teacher and are not bolded. [†]Completion-only masking yields near-chance WinoGrande and HellaSwag *generation* accuracy at 1.7B (extraction artifact, Appendix M); the corresponding perplexity-based scores are intact.

| Model | Knowledge | | | Common Sense | | Reasoning | | | | | IF |
|---|---|---|---|---|---|---|---|---|---|---|---|
| | C-Eval | MMLU-R | CMMLU | WG | HS | ARC-E | ARC-C | BBH | GSM8K | HEval | IFEval |
| *Teacher (Qwen3-0.6B)* | 42.9 | 46.1 | 45.2 | 51.2 | 39.0 | 70.8 | 55.0 | 27.4 | 57.5 | 35.3 | 57.1 |
| Pure KDA | 26.2 | 29.4 | 35.1 | 38.6 | 20.3 | 47.9 | 33.9 | 5.4 | 22.8 | 12.8 | 34.7 |
| Pure Mamba | 17.9 | 18.0 | 25.2 | 24.7 | 17.3 | 20.7 | 19.9 | 8.4 | 8.3 | 6.1 | 28.2 |
| Hybrid Mamba | 37.8 | 38.6 | 37.1 | 50.1 | 29.0 | 63.3 | 46.2 | 12.1 | 28.3 | 11.2 | 43.7 |
| Hybrid KDA | 38.2 | **40.3** | **38.7** | **50.7** | **33.9** | **63.5** | **48.4** | **18.9** | **34.5** | **19.9** | **46.7** |
| Hybrid GDN | 32.5 | 27.3 | 27.9 | 29.5 | 14.2 | 39.7 | 26.7 | 12.0 | 20.3 | 10.1 | 38.9 |
| Hybrid GLA | **38.8** | 28.2 | 25.9 | 44.6 | 18.8 | 51.0 | 37.7 | – | 19.0 | 1.5 | 33.6 |
| Hybrid Lightning | 18.6 | 10.2 | 12.8 | 43.4 | 11.3 | 14.9 | 9.6 | – | 9.6 | 0.0 | 15.7 |
| *Teacher (Qwen3-1.7B)* | 62.8 | 66.9 | 62.0 | 54.2 | 59.8 | 84.8 | 73.6 | 34.3 | 80.9 | 58.5 | 66.1 |
| Hybrid KDA (1.7B) | 56.5 | 55.0 | 54.0 | 33.2[†] | 23.5[†] | 68.5 | 58.2 | 21.2 | 60.1 | 34.8 | 58.6 |

"–": BBH not evaluated for Hybrid GLA and Hybrid Lightning.
The "Mamba" rows use a Mamba2 mixer with the same architectural modifications as the original MOHAWK distillation (Bick et al., 2024).

Table 4 shows that retaining a subset of attention layers is critical: Hybrid models consistently outperform their Pure counterparts, with the largest separations on multi-step reasoning benchmarks (BBH, GSM8K). This supports preserving attention for compositional computation while using linear mixers for the majority of layers. Among the linear mixers, KDA is the strongest, consistently outperforming Mamba2, with the advantage most pronounced on the hardest reasoning and code-generation tasks. Every mixer here is distilled through the identical GenDistill pipeline, so the comparison isolates the sequence mixer. Because Mamba2 is the student used by MOHAWK and Llamba, the Mamba rows act as a controlled prior-student baseline at the same teacher, token budget, and data. The three additional mixers trail KDA and Hybrid Mamba on average and trade off against each other across task categories: GLA is stronger on knowledge and common-sense benchmarks, whereas Gated DeltaNet is stronger on math and code generation. Lightning Attention

is the weakest of the sequence mixers, failing to reach a usable generation regime under the shared recipe[7]. Perplexity-based results for all architectures are reported in Appendix O (Table 22).

Overall, the best Hybrid-KDA student retains 86–99% of teacher accuracy on knowledge and common-sense tasks, confirming that the majority of the teacher's factual and linguistic knowledge transfers successfully. The same architecture and recipe scale to a Qwen3-1.7B teacher (Table 4, lower block), where Hybrid-KDA again tracks its teacher across knowledge, reasoning, code, and instruction following, with only the commonsense multiple-choice benchmarks depressed by the completion-only extraction artifact noted in the table. The largest remaining gaps appear on multi-step reasoning (BBH: 69%, GSM8K: 60%) and code generation (HumanEval: 56%), tasks that require extended autoregressive coherence. Closing these gaps likely requires either larger token budgets, on-policy distillation, or dedicated RL training.

### 5.3.1 Efficiency Analysis

Having selected the architecture on quality, we now turn to the efficiency that motivates replacing attention in the first place, since linear-time mixers avoid the quadratic compute and growing KV cache that attention incurs at long context.

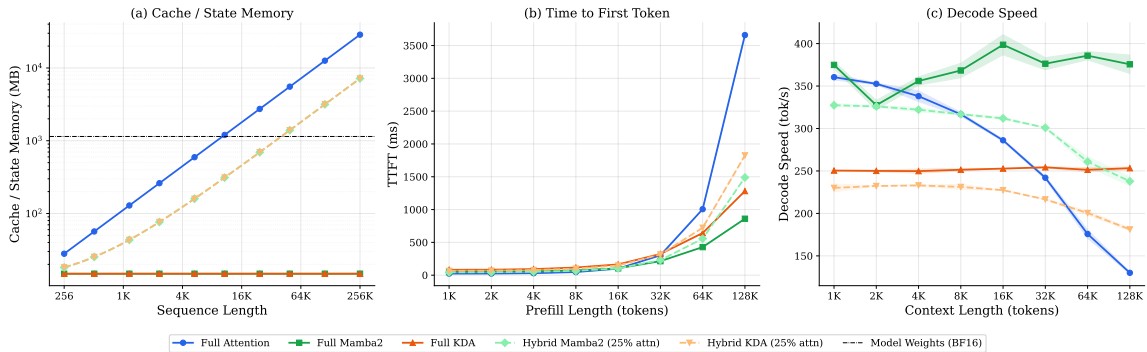

Figure 4: Inference efficiency comparison (single GPU, batch size 1). **(a)** Memory usage: SSMs use constant $\mathcal{O}(1)$ state, reducing hybrid memory footprint by up to 75%. **(b)** Time to First Token: Linear SSM scaling yields 2–4× speedups at long contexts (128K) despite short-context overhead. **(c)** Decode speed: SSMs maintain constant throughput; KDA reaches parity with Transformers at 32K tokens.

Figure 4 characterizes inference efficiency. Hybrid architectures reduce KV cache memory by up to 75%[8] compared to Transformers, enabling 4× larger batch sizes or sequence lengths. While SSM prefill ($\mathcal{O}(n)$) incurs kernel overhead at short sequences, it scales linearly, becoming 2–4× faster than attention ($\mathcal{O}(n^2)$) at 128K tokens. Decoding throughput is stable; KDA is ∼30% slower than Transformers at short contexts but reaches parity by 32K tokens. Throughput under GPU saturation (sweeping batch sizes to OOM) is reported in Appendix P; at 128K tokens, hybrid models achieve up to 5× the teacher's peak throughput by fitting larger batches.

## 6 Discussion

Our results show that generation quality in cross-architecture distillation depends on both the student architecture and the distillation recipe, and that evaluating models by generation is necessary to judge either one reliably. In our study, perplexity and generation accuracy tell different stories on the design choices that most affect the student, namely the training objective (KD vs. SFT), the loss-masking scope, attention freezing (Table 2), and the choice of sequence mixer (Section 5.3). SFT achieves *lower* perplexity than KD

---

[7]Lightning Attention's weak performance may stem from the shared GenDistill recipe not being tuned to this specific mixer, although the same untuned recipe is applied to all the other mixers as well.

[8]Replacing attention with KDA or Mamba2 slightly *increases* total parameter count (613M vs. 596M non-embedding) due to extra gating and convolution parameters; the efficiency gains stem from $\mathcal{O}(n)$ vs. $\mathcal{O}(n^2)$ complexity scaling, not parameter reduction.

but *worse* generation accuracy by 8–11 pp, and freezing attention layers yields a 5.2 pp gain in generation accuracy with less than 1 pp difference in perplexity. The same disagreement persists when we scale the student to a 1.7B teacher (Section 5.1), and it also appears when evaluating a public distilled checkpoint, where QRWKV6-7B-Instruct nearly matches its Qwen2.5-7B teacher under perplexity but trails by 20.8 pp under generation-based evaluation (Table 5). Although the limitations of perplexity as a quality proxy are well known, our results suggest that the problem is particularly severe in cross-architecture distillation, where perplexity-only evaluation is still common. We therefore argue that generation-based evaluation should be the primary protocol for distilled models in this setting.

With this evaluation lens established, our ablations identify three critical design choices for cross-architecture distillation at the sub-1B scale. First, **use KD with completion-only masking**: Table 17 shows that it dominates on generation benchmarks across knowledge, reasoning, and instruction-following tasks, while SFT only appears competitive under perplexity-based evaluation. Second, **use teacher-aligned instruction data**: Table 3 shows a 20 pp swing in average accuracy from dataset choice alone, with teacher-aligned data (AM-Qwen3-Distilled) outperforming broader instruction corpora by a wide margin. Third, **freeze attention layers during Stage 3b**: Table 2 shows a 5.2 pp average generation improvement from freezing, with minimal effect under perplexity-based evaluation. These choices share a common logic, each preserving the teacher's pretrained structure and steering the student's limited capacity toward generation rather than letting instruction tuning overwrite what pretraining established.

Two further choices are less clear-cut. Extending Stage 3b training is a trade-off rather than a uniform gain, improving Chinese knowledge and instruction following while degrading some English benchmarks (Table 17), so the best duration depends on the target evaluation profile. The pipeline structure itself also matters, as merging Stages 3a and 3b into a single interleaved phase underperforms the sequential pipeline at matched compute (Appendix L), confirming the value of staged training with dedicated objectives.

The architecture comparison points to a single best student, Hybrid-KDA, but its value is conditional on the deployment regime. The efficiency that motivates replacing attention only materializes at long context, where the hybrid wins decisively on memory, time-to-first-token, and decode throughput (Figure 4), whereas at short context it is slower than the Transformer teacher it replaces. The hybrid is therefore best understood as a targeted substitute for long-context workloads rather than a universal drop-in.

## 6.1 Limitations

Comparing whole distillation pipelines is difficult, since published pipelines differ along many axes at once (e.g. the teacher, the student mixer, the number of stages, the loss functions, token budgets, datasets). These axes are also interdependent. A head-to-head between two pipelines then shows which bundle of choices wins on a given setup, and cannot isolate the ingredient responsible. Consequently, our aim is not to make concrete claims that GenDistill produces higher-quality models than other distillation pipelines. Our contribution is the generation-based evaluation analysis and the resulting recipe, a controlled attribution of which distillation ingredients drive generation quality.

A further limitation is long-context *quality*, as distinct from long-context *efficiency*. The hybrid wins decisively on efficiency (Figure 4), but distillation on 4K-token sequences does not transfer to longer contexts: Hybrid KDA retains only 55% of teacher LongBench accuracy and degrades on NIAH retrieval beyond 16K tokens (Appendix Q). This reflects the missing long-context training stage rather than an architectural limit, since the teacher itself acquired its 32K capability through a dedicated phase (Yang et al., 2025a), and concurrent work recovers long-context quality once such a stage is added (Chen et al., 2026; Team et al., 2026).

Finally, our Stage 3b training used only 250M–750M tokens from static datasets. On-policy distillation using student-generated responses (Agarwal et al., 2024) and larger compute budgets could close the remaining quality gap. Dataset composition effects such as curriculum ordering and mixing ratios were also not explored and may yield further improvements.

# 7 Conclusion

This work shows that producing distilled hybrid models with genuine generation quality requires jointly designing the student architecture and the distillation pipeline, guided by generation-based evaluation. Perplexity-based metrics, still the norm in cross-architecture distillation, are a systematically misleading proxy. The divergence appeared consistently across every design axis in our 0.6B-parameter ablations, persisted in a 1.7B scale-up (Section 5.1), and is corroborated at 7B scale (Table 5). While the general limitation of perplexity is well-known, our results show this gap is especially severe in cross-architecture distillation, where perplexity-only reporting can lead to incorrect conclusions about which methods actually produce models that generate well. We argue that generation-based evaluation should be the primary protocol in this setting.

On the practical side, GenDistill's systematic ablations yield a concrete distillation recipe: KD with completion-only masking, teacher-aligned instruction data, and frozen attention layers. This combination produces a Hybrid-KDA student that retains 86–90% of teacher accuracy on knowledge benchmarks while reducing KV cache memory by ∼75%. Confirming the recipe at multi-billion-parameter scale, and extending it with dedicated long-context training, remain the most pressing directions for future work.

## Broader Impact Statement

By lowering the compute and memory cost of running capable language models, cross-architecture distillation can broaden access to these models and enable deployment at larger scale and on more constrained hardware. At the same time, cheaper and faster generation lowers the cost of producing text at scale, so the standard misuse considerations for large language models, such as large-scale generation of low-quality or misleading content, apply here as well, with efficiency changing the scale of these risks rather than their qualitative nature. Beyond this, we do not foresee risks specific to this work.

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

This appendix collects the derivations, full result tables, and ablations that support the main text, organized into the thematic groups below.

## A    Motivating Example, Evaluation Protocol Gap

This section gives the full numbers behind the motivating example shown in the introduction (Figure 1).

Table 5: Evaluation protocol gap for QRWKV6-7B-Instruct, distilled from Qwen2.5-7B-Instruct. PPL-based results are taken from the original paper; generation-based results are our evaluation using EvalScope. Both columns evaluate the *same model checkpoint*; only the evaluation protocol differs. On overlapping benchmarks, the average teacher–student gap jumps from **0.2 pp** (PPL) to **20.8 pp** (generation).

| | PPL-based Evaluation | | | | Generation-based Evaluation | | | | | | |
|---|---|---|---|---|---|---|---|---|---|---|---|
| Model | ARC-E | ARC-C | HS | WG | ARC-E | ARC-C | HS | WG | GSM8K | HEval | IFEval |
| Qwen2.5-7B-Instruct | 81.5 | 55.0 | 80.5 | 71.4 | 91.9 | 87.7 | 82.3 | 66.2 | 90.2 | 85.4 | 71.4 |
| QRWKV6-7B-Instruct | 81.4 | 56.3 | 79.0 | 71.1 | 75.2 | 62.4 | 49.9 | 57.5 | 17.4 | 31.7 | 46.6 |

## B    Kimi Delta Attention Formulation

**Linear attention as associative memory.** Linear attention (Katharopoulos et al., 2020) replaces the softmax attention kernel with an outer-product update to a matrix-valued recurrent state $\mathbf{S}_t \in \mathbb{R}^{d_k \times d_v}$:

$$\mathbf{S}_t = \mathbf{S}_{t-1} + \boldsymbol{k}_t \boldsymbol{v}_t^\top, \qquad \boldsymbol{o}_t = \mathbf{S}_t^\top \boldsymbol{q}_t. \tag{1}$$

$\mathbf{S}_t$ serves as an associative memory that stores key–value associations. However, the purely additive update accumulates information without any forgetting, causing interference as the context grows.

**The delta rule and Gated DeltaNet.** DeltaNet (Schlag et al., 2021; Yang et al., 2024) addresses this by interpreting the state update as online gradient descent on a per-token reconstruction loss $\mathcal{L}_t(\mathbf{S}) = \frac{1}{2}\|\mathbf{S}^\top \boldsymbol{k}_t - \boldsymbol{v}_t\|^2$, yielding a rank-1 corrective update:

$$\mathbf{S}_t = (\mathbf{I} - \beta_t \boldsymbol{k}_t \boldsymbol{k}_t^\top)\,\mathbf{S}_{t-1} + \beta_t \boldsymbol{k}_t \boldsymbol{v}_t^\top, \tag{2}$$

where $\beta_t$ is a learned step size. This delta rule enables the memory to correct stale associations rather than merely accumulating them. Gated DeltaNet (GDN) (Yang et al., 2025c) further introduces a *scalar* forget gate $\alpha_t \in [0, 1]$, acting as weight decay on the memory:

$$\mathbf{S}_t = \alpha_t\,(\mathbf{I} - \beta_t \boldsymbol{k}_t \boldsymbol{k}_t^\top)\,\mathbf{S}_{t-1} + \beta_t \boldsymbol{k}_t \boldsymbol{v}_t^\top. \tag{3}$$

This provides a principled mechanism to control memory lifespan. However, GDN's scalar gate applies a uniform decay rate across all feature dimensions, limiting the model's ability to selectively retain or forget information at a fine-grained level.

**Channel-wise gating in KDA.** Kimi Delta Attention (KDA) (Team et al., 2025) refines GDN by replacing the scalar gate with a *channel-wise* (i.e., per-dimension) diagonal gate $\mathrm{Diag}(\boldsymbol{\alpha}_t) \in \mathbb{R}^{d_k \times d_k}$, where each feature dimension maintains an independent forgetting rate:

$$\mathbf{S}_t = (\mathbf{I} - \beta_t \boldsymbol{k}_t \boldsymbol{k}_t^\top)\,\mathrm{Diag}(\boldsymbol{\alpha}_t)\,\mathbf{S}_{t-1} + \beta_t \boldsymbol{k}_t \boldsymbol{v}_t^\top, \qquad \boldsymbol{o}_t = \mathbf{S}_t^\top \boldsymbol{q}_t. \tag{4}$$

## C  Model Architecture Details

**Base model (teacher).** We distill from Qwen3-0.6B, a dense Transformer with grouped-query attention. Table 6 summarizes its configuration.

Table 6: Qwen3-0.6B base model (teacher) configuration.

| Parameter | Value |
|---|---|
| Hidden dimension ($d_{\mathrm{model}}$) | 1024 |
| Number of layers | 28 |
| Attention heads (Q) | 16 |
| KV heads (GQA) | 8 |
| Head dimension | 128 |
| Intermediate size (MLP) | 3072 |
| Vocabulary size | 151,936 |
| Max position embeddings | 40,960 |
| RoPE $\theta$ | 1,000,000 |
| RMSNorm $\epsilon$ | 1e-6 |
| MLP activation | SiLU |
| Tie word embeddings | Yes |

The student models are based on the 596M-parameter teacher (non-embedding) but replace a subset of attention layers with alternative sequence mixers. Each mixer layer introduces additional gating, convolution, and state-management parameters beyond the original Q/K/V/O projections, increasing the total parameter count: KDA adds ∼0.8M per replaced layer (short convolutions on Q/K/V, gating projections for the delta rule), while Mamba2 adds ∼2.1M per replaced layer (a larger joint input projection that maps to the SSM state components, time-step, and gating dimensions). Table 7 details the mixer configurations. Each mixer layer replaces the self-attention block while retaining the original MLP, layer norms, and residual connections. Pure KDA and Mamba variants use identical mixer configurations but replace all 28 layers.

Table 7: Hybrid student mixer configurations (Qwen3-0.6B scale). KDA = Kimi Delta Attention.

| Parameter | KDA | Mamba2 |
|---|---|---|
| Retained attention layers | 7 of 28 | 7 of 28 |
| Attn. layer indices | {0,2,6,11,13,18,21} | {0,2,6,11,13,18,21} |
| State dimension ($d_{\text{state}}$) | 128 per head | 128 per head |
| Q heads | 16 | 16 |
| K/V heads (GQA) | 8 | 8 |
| Head dimension (Q/K/V) | 128 / 128 / 128 | 128 / 128 / 128 |
| Chunk size | 128 | 128 |
| Short convolution kernel | 4 | 4 |
| Output gate activation | SiLU | SiLU |
| Post-SSM normalization | No | No |
| QK normalization (RMSNorm) | Yes | No |
| Initialization | VO[a] | VO[a] |
| Non-embedding parameters | 613M | 641M |

[a]Teacher's V and O projection weights used to initialize the corresponding mixer parameters.

## D   Hyperparameter Details

Table 8 lists the full hyperparameter configuration for each distillation stage. Stage 1 aligns the outputs of the attention layer between the two models, while Stage 2 aligns the outputs of the whole attention block via L2 loss on pretraining data. Stage 1 only trains the Q and K projections, keeping the conv1d (identity initialized) and other parameters fixed. Stage 2, meanwhile, trains all mixer parameters. Stages 3a and 3b perform end-to-end knowledge distillation with forward KL divergence. Both stages freeze the attention layers to prevent catastrophic forgetting of the representations inherited from the teacher model. Stage 3b switches to instruction data with completion-only masking (see Section 5.1). All stages use AdamW with $(\beta_1, \beta_2) = (0.9, 0.95)$, a warmup-stable-decay learning rate schedule (10% warmup, 10% decay, minimum LR ratio 0.02), weight decay 0.1, gradient clipping at 1.0, bf16 mixed precision, and DeepSpeed ZeRO Stage 2.

**Evaluation configurations.**   We use two complementary evaluation frameworks, chosen to highlight the PPL-vs-generation gap that is a central finding of this work. All evaluations use a batch size of 64 and the SDPA (Scaled Dot-Product Attention) backend. *EvalScope (generation-based):* Models are evaluated using the recommended sampling setup: temperature=0.7, top-$p$=0.8, top-$k$=20, and seed=42. Maximum new tokens is set to 1024. Few-shot settings: C-Eval (5-shot), MMLU / MMLU-Redux (5-shot), ARC-Easy / ARC-Challenge (0-shot), HellaSwag (0-shot), WinoGrande (0-shot), GSM8K (4-shot chain-of-thought), CMMLU (0-shot), BBH (0-shot), HumanEval (0-shot), IFEval (0-shot), TruthfulQA (0-shot). *LM-Eval Harness (perplexity-based):* Models are evaluated on likelihood-based scoring. Tasks: ARC-Easy, ARC-Challenge, HellaSwag, WinoGrande, PIQA, LAMBADA (OpenAI), MMLU. We use 3 seeds for the evals, seeds = 1, 2, 3.

Table 8: Hyperparameters for each distillation stage.

| Hyperparameter | Stage 1 *(Mixer alignment)* | Stage 2 *(Hidden-state align.)* | Stage 3a *(End-to-end KD)* | Stage 3b *(Instruction KD)* |
|---|---|---|---|---|
| *Data* | | | | |
| Dataset | FineWeb-EDU + FW-EDU-Chinese | FineWeb-EDU + FW-EDU-Chinese | FineWeb-EDU + FW-EDU-Chinese | AM-Qwen3-Distilled (completion-only, packed) |
| Tokens | 30M | 30M | 500M | 500M |
| Sequence length | 2048 | 2048 | 2048 | 4096 |
| *Optimization* | | | | |
| Learning rate | 5e-4 | 1e-4 | 2.5e-5 | 2.5e-5 |
| Per-device batch size | 8 | 8 | 2 | 2 |
| Gradient accumulation | 1 | 1 | 16 | 16 |
| Effective batch size (2 GPUs) | 16 | 16 | 64 | 64 |
| *Loss* | | | | |
| Training objective | L2 norm (mixer layer output align.) | L2 norm (attention block output align.) | Forward KL divergence | Forward KL divergence |
| KD temperature $\tau$ | — | — | 1.0 | 1.0 |
| CE loss weight $\alpha_{\mathrm{CE}}$ | — | — | 0.0 | 0.0 |
| Top-$K$ | — | — | None (full vocab) | None (full vocab) |
| *Parameter Freezing* | | | | |
| Frozen parameters | None | None | Attention layers | Attention layers |
| MLP active | No | Yes | Yes | Yes |

# E   Attention Layer Selection Ablation

The main text (Section 3.1.1) separates this within-pipeline use of perplexity from the final-evaluation use the paper argues against, and we expand on that distinction here. The failure mode the paper documents concerns a fully trained student scored over completion tokens, where the teacher–student likelihood gap is what perplexity is asked to capture. Here perplexity instead scores an undertrained scaffold whose role is exploration rather than deployment, so neither the artefact measured nor the quantity it stands for is the same.

The design space for layer selection has four axes: the probe dataset, the search procedure, the scoring metric, and the amount of student-side training performed before each measurement. The principled upper bound on this spectrum is the attention-placement step of PostNAS (Gu et al., 2025), which trains a once-for-all super-network containing both attention and linear pathways and then runs beam search over its sub-networks using direct downstream-task signals such as loss on the correct MMLU answer span and task accuracy on math and retrieval. PostNAS is generation-aware in both probe and metric, and consistent with the broader thesis of this paper. It is also expensive: per its Appendix A.2, the attention-placement step consumes 50B tokens and ~800 H100 GPU-hours. Two more compute-efficient points on this spectrum trade per-candidate training cost for proxy quality. GA-S2 (Li et al., 2025) performs a single-replacement probe at every layer and scores each variant by teacher and student KL after a short distillation pass per variant (~700M tokens per probe, totalling between ~7 and ~26B tokens at our scale). Our procedure removes per-candidate training entirely. We distill a fully-linear scaffold once at 100M tokens, then run beam search over attention-layer additions and score each candidate by held-out perplexity averaged across three probes: associative recall, a key/value retrieval task in the same spirit as PostNAS's KV signal; C-EVAL, a knowledge benchmark; and FineWeb-EDU, a generic language-modelling probe. Two of the three probes are task-shaped, so the input side of our signal is closer in form to PostNAS-style task-aware evaluation than to a generic LM-PPL baseline.

We compare beam search layer selection against three baselines for the attention budget of 5 attention layers. We choose this, as with increasing number of attention layers the performance will converge to the teacher model's performance and the gap between methods will therefore be smaller. We compare the following selection methods:

- **Uniform:** Layers are spaced at equal intervals.

- **Greedy:** Selects top-$k$ layers by individual perplexity impact.

- **Greedy Aggregate (Learned):** Starts from a linear model and adds attention layers based on held-out KL divergence after training the resulting hybrid models (Li et al., 2025).

- **Beam Search Add:** Starts from a full KDA model and adds attention layers to maximize performance.

- **Beam Search Replace:** starts with a full transformer and progressively replaces attention layers with KDA layers, minimizing perplexity degradation.

**Evaluation Tasks.** We consider evaluating the layer importance on three tasks, and use the average performance on all 3 tasks as the importance measure:

- **AR (Associative Recall):** A synthetic key-value retrieval task that directly tests in-context retrieval capabilities. We use 40 key-value pairs per example with 500 total examples.

- **CEVAL:** Chinese language understanding benchmark (similar task to MMLU, whose performance has been shown to be highly influenced by the choice of attention layers (Bick et al., 2025b)).

- **Fineweb-EDU:** General language modeling perplexity on educational web text.

Table 9 lists selected indices. Table 10 reports downstream performance. Greedy Aggregate achieves the lowest evaluation loss (Figure 5) but underperforms on downstream tasks. Beam Add achieves the best balance of reasoning and knowledge (Table 10), outperforming both Beam Replace and Greedy Learned. We therefore use the attention layers selected by the Beam Search Add method in all our remaining experiments.

Interestingly, the method with the lowest in-pipeline evaluation loss (Greedy Learned / GA-S2) is not the one with the strongest downstream performance (Table 10 and Figure 5), which is itself an instance of the perplexity and generation disagreement this paper documents, here surfacing inside the layer-selection step rather than at final evaluation. We hypothesise that two choices in our procedure compensate for using a weaker per-candidate metric than KL. Beam search explores beyond locally greedy choices, and the three-probe ensemble averages across capacity-shaped, retrieval-shaped, and knowledge-shaped signals. Together they avoid the $\sim100\times$ search-stage compute overhead of GA-S2's per-layer retraining without degrading selection quality on aggregate downstream. With PostNAS-level compute a direct downstream-task signal would still be preferable. In the budget regime our pipeline targets, beam search over probe-PPL is the most compute-efficient point on the spectrum that does not degrade selection quality.

Table 9: Attention layer selection for different methods and sparsity levels (Qwen3-0.6B, 28 layers). Layer indices are 0-indexed.

| Method | 5 Attn Layers | 7 Attn Layers | 9 Attn Layers |
|---|---|---|---|
| Uniform | {0,6,11,16,21} | {0,5,9,13,17,21,25} | {0,3,6,9,12,15,18,21,24} |
| Learned Selection (Greedy Aggregate, KL) | {6,16,19,20,21} | {6,8,11,16,19,20,21} | {2,6,8,11,16,18,19,20,21} |
| Greedy | {0,2,11,13,18} | {0,2,5,11,13,16,18} | {0,1,2,5,7,11,13,16,18} |
| Beam Search Add | {2,11,12,17,21} | {0,2,6,11,13,18,21} | {0,1,2,6,11,17,18,19,21} |
| Beam Search Replace | {0,3,12,14,23} | {0,3,12,14,18,23,24} | {0,3,7,12,14,18,23,24,25} |

Table 10: Downstream performance after Stage 3b for the selected attention-layer configurations.

| Config (Stage 3b) | CEVAL ↑ | ARC-C ↑ | ARC-E ↑ | WG ↑ | HS ↑ | PIQA ↑ | MMLU-redux ↑ |
|---|---|---|---|---|---|---|---|
| Uniform | 35.1 | 42.8 | 56.1 | 44.5 | 21.2 | 64.0 | 36.1 |
| Greedy Learned | 30.3 | 44.9 | **60.1** | 49.5 | **33.5** | **64.6** | **38.7** |
| Beam Replace | 26.8 | 37.3 | 49.2 | **51.2** | 26.6 | 63.1 | 32.0 |
| Beam Add | **36.0** | **47.0** | 59.2 | 50.2 | 33.1 | 64.1 | 37.1 |
| Greedy | 32.3 | 20.9 | 30.3 | 31.4 | 12.3 | 64.2 | 27.3 |

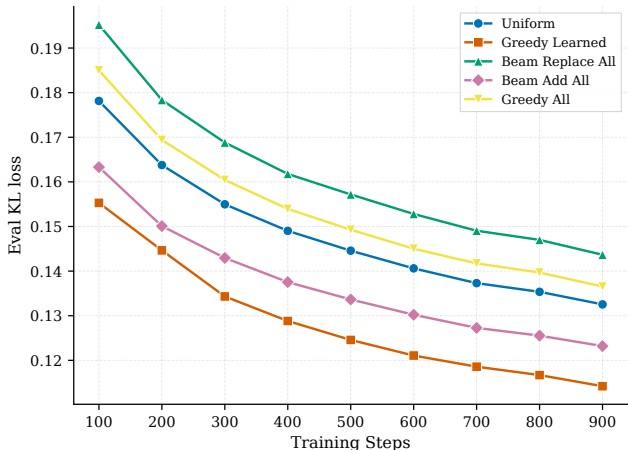

Figure 5: Evaluation loss curves for attention layer selection methods during Stage 3b. Greedy Learned achieves the lowest loss but Beam Add yields better downstream performance.

## F    Architecture Design Choices

We conduct an ablation study to analyze the impact of key architectural choices on the distilled hybrid model. Specifically, we investigate: (i) the output gate activation function (SiLU vs. Sigmoid), (ii) the presence of a post-SSM normalization layer, and (iii) the initialization strategy for the mixer parameters (VO initialization from the teacher's value and output projections, QKVO initialization from all attention projections, or random initialization). For each configuration, we run the three-stage distillation pipeline on the FineWeb-Edu dataset (for stage 3 we do not train on the instruct data, but only the 100M tokens on the FineWeb-Edu) and report the evaluation perplexity at the end of training.

Table 11: Ablation study on model architecture details. We report evaluation perplexity on FineWeb-Edu after completing Stages 1–3 of distillation.

| Configuration | Gate | Post-SSM Norm | Initialization | Eval PPL ↓ |
|---|---|---|---|---|
| Baseline (Ours) | SiLU | ✗ | VO | 21.32 |
| Sigmoid gate | Sigmoid | ✗ | VO | 21.64 |
| Post-SSM norm | SiLU | ✓ | VO | 21.47 |
| QKVO init | SiLU | ✗ | QKVO | 21.25 |
| QKVO init (no Stage 1) | SiLU | ✗ | QKVO | 25.41 |
| VO init (no Stage 1) | SiLU | ✗ | VO | 24.42 |
| Random init | SiLU | ✗ | Random | 24.88 |

We find Stage 1 to be critical for stable and strong distillation: skipping Stage 1 significantly degrades performance, even when initializing from the teacher, while fully random initialization remains competitive but worse overall, consistent with observations in prior attention-to-linear distillation work (Goldstein et al., 2025). Using a Sigmoid gate underperforms SiLU. We hypothesize this is partly because Sigmoid does not admit an identity-style initialization for the gate during distillation, which may hinder optimization. Adding a post-SSM normalization layer is slightly worse in perplexity and, in our downstream evaluation, shows a larger drop and we therefore omit it (and we also observed it to be substantially worse in our Mamba-based hybrids). Finally, QKVO initialization performs similarly (slightly better for the KDA models) than VO-only initialization, however, in early Mamba-based experiments the gap was larger in the opposite direction, so we default to VO initialization for a fairer comparison across architectures.

## G  Top-K Sparsification and CE Loss Weight

We ablate top-$K$ sparsification of the KD target distribution and the cross-entropy loss weight $\alpha_{\text{CE}}$. Ta-

Table 12: Effect of top-$K$ sparsification of the KD loss during Stage 3b with frozen attention. All runs use 7 attention layers, `freeze_attn=True`, KD with completion-only masking, and 250M tokens. Generation-based evaluation (EvalScope). Best result per column in **bold**.

| Top-$K$ | C-Eval | MMLU-R | CMMLU | ARC-E | ARC-C | HS | WG | BBH | GSM8K | HEval | IFEval | Avg. |
|---------|--------|--------|-------|-------|-------|-----|-----|-----|-------|-------|--------|------|
| None (baseline) | **39.4** | **39.3** | 37.7 | 63.0 | **50.9** | 36.4 | 49.6 | **19.2** | **35.7** | 17.1 | **45.5** | **39.4** |
| $K = 100$ | 34.9 | 38.3 | 38.0 | 63.8 | **50.9** | 35.6 | 49.6 | 18.3 | 35.1 | **20.1** | 43.8 | 38.9 |
| $K = 500$ | 35.9 | 39.0 | 37.0 | 63.1 | 49.5 | **36.4** | 48.8 | 18.8 | 32.6 | 18.3 | 44.4 | 38.5 |
| $K = 1,000$ | 34.9 | 38.8 | 37.7 | **63.9** | 50.3 | 36.5 | 50.4 | 18.4 | 33.6 | 19.5 | 44.9 | 39.0 |
| $K = 10,000$ | 35.8 | 38.7 | **38.1** | 63.7 | 49.9 | 36.1 | **51.1** | 18.1 | 32.4 | 20.7 | 45.3 | 39.1 |

ble 12 shows that restricting the KD loss to the teacher's top-$K$ tokens does not improve performance when attention layers are frozen. The full-vocabulary baseline achieves the highest average score across all eleven benchmarks, with the largest advantages on knowledge-intensive tasks (C-Eval, MMLU-Redux, GSM8K). Commonsense benchmarks show no systematic trend across $K$ values. We therefore use full-vocabulary KD for all main experiments. Adding any cross-entropy weight substantially degrades knowledge benchmarks

Table 13: Effect of cross-entropy loss weight $\alpha_{\text{CE}}$ in the KD objective ($\mathcal{L} = \mathcal{L}_{\text{KL}} + \alpha_{\text{CE}}\mathcal{L}_{\text{CE}}$). All runs use 5 attention layers, `freeze_attn=False`, 500M tokens.

| $\alpha_{\text{CE}}$ | WG | HS | ARC-E | ARC-C | C-Eval | MMLU-R |
|---------|------|------|-------|-------|--------|--------|
| Default | 40.7 | 17.5 | 50.0 | 37.7 | **35.7** | **35.8** |
| 0.1 | 40.8 | **18.3** | 44.0 | 31.1 | 22.3 | 28.3 |
| 0.035 | **44.0** | 16.7 | 48.8 | 36.7 | 22.9 | 33.5 |
| 0.01 | 43.4 | 17.0 | **51.2** | **38.2** | 27.3 | 34.4 |

(Table 13): even $\alpha_{\text{CE}} = 0.01$ drops C-Eval by 8.4 points and MMLU-R by 1.4 points, while higher values cause catastrophic drops. We accordingly set $\alpha_{\text{CE}} = 0.0$ for all main experiments.

## H  Learning Rate Ablation

Several generation benchmarks (HellaSwag, ARC-E, MMLU-Redux) degrade as Stage 3b training continues at the default LR=2.5e-5 (Section 5.1). This appendix and the next isolate the cause. Here we test whether a smaller learning rate (LR=1e-5) mitigates the effect by slowing the update dynamics. Appendix I instead tests whether the MLP parameters are responsible. All other settings are held fixed (KD with completion-only masking) and we vary the training token budget. Table 14 shows that a modest learning-rate reduction does not remove the downward trend on several generation metrics with additional Stage 3b tokens: ARC-E and HellaSwag decrease as training continues under both LR settings. In contrast, it slows improvement on knowledge-heavy benchmarks at a fixed token budget: for example, C-Eval at 250M tokens is substantially lower at LR=1e-5 than at LR=2.5e-5. Consequently, we stay with the chosen LR. Note that PPL-based evaluation changes little with learning rate, while differences in generation-based metrics are substantial.

Table 14: Learning rate ablation for Stage 3b training with KD loss (KD) and completion-only masking (C). Best student result per column and evaluation method in **bold**.

| | Tokens | LR | WG ↑ | HS ↑ | ARC-E ↑ | ARC-C ↑ | C-Eval ↑ | MMLU-R ↑ |
|---|---|---|---|---|---|---|---|---|
| Generation | Teacher (Qwen3-0.6B) | | 51.2 | 39.0 | 70.8 | 55.0 | 42.9 | 46.1 |
| | 250M | 2.5e-5 | 48.9 | 35.4 | **65.6** | 48.8 | 37.9 | **41.6** |
| | 500M | 2.5e-5 | 48.1 | 33.9 | 64.1 | **50.0** | 39.9 | 40.7 |
| | 750M | 2.5e-5 | **49.4** | 32.3 | 62.3 | 47.0 | **41.0** | 39.0 |
| | 100M | 1e-5 | 49.8 | 34.0 | 64.8 | 49.1 | 28.8 | 40.7 |
| | 250M | 1e-5 | 48.9 | **35.5** | 64.2 | 47.0 | 31.3 | 41.5 |
| | 500M | 1e-5 | 48.6 | 34.7 | 61.4 | 48.2 | 36.9 | 39.0 |

| | Tokens | LR | WG ↑ | HS ↑ | ARC-E ↑ | ARC-C ↑ | PIQA ↑ | LAMBADA ↓ |
|---|---|---|---|---|---|---|---|---|
| PPL-based | Teacher (Qwen3-0.6B) | | 55.5 | 47.3 | 55.6 | 34.2 | 65.9 | 24.7 |
| | 250M | 2.5e-5 | 56.4 | 44.9 | 54.9 | 31.7 | 65.2 | 40.6 |
| | 500M | 2.5e-5 | 55.8 | 45.0 | 54.8 | 31.1 | 65.1 | 36.5 |
| | 750M | 2.5e-5 | 56.3 | 45.0 | 54.9 | 32.3 | 65.0 | 37.1 |
| | 100M | 1e-5 | **57.1** | 44.8 | 53.2 | 31.1 | 65.4 | **34.2** |
| | 250M | 1e-5 | 55.2 | 44.9 | 54.1 | 31.9 | **65.5** | 36.5 |
| | 500M | 1e-5 | 55.4 | **44.9** | **54.7** | **31.9** | **65.7** | 37.8 |

# I  Frozen MLP Ablation

Section 5.1 shows that several generation benchmarks degrade with extended Stage 3b training and the learning-rate ablation (Appendix H) rules out learning rate as the primary explanation. Here we test whether the degradation is driven by corruption of factual knowledge stored in MLP parameters by freezing all MLP layers during Stage 3b while keeping all other settings fixed (KD with completion-only masking). Unfrozen KD baselines from Table 17 are reproduced for comparison. Freezing the MLPs does not eliminate the

Table 15: Frozen MLP ablation across training durations. Generation-based (EvalScope, left) and perplexity-based (LM-eval, right) results for Stage 3b with frozen vs. unfrozen MLP parameters under KD with completion-only masking. Unfrozen rows are reproduced from Table 17. HS generation is unavailable for frozen runs[10]. Best student result per column in **bold**.

| MLP | Tokens | WG | HS | ARC-E | ARC-C | C-Eval | MMLU-R | CMMLU | GSM8K | HEval | BBH | IFEval | HS | PIQA | LAM.↓ | MMLU |
|---|---|---|---|---|---|---|---|---|---|---|---|---|---|---|---|---|
| | | | | | | Generation-based (EvalScope) | | | | | | | | Perplexity-based (LM-eval) | | |
| Teacher (Qwen3-0.6B) | | 51.2 | 39.0 | 70.8 | 55.0 | 42.9 | 46.1 | 45.2 | 57.5 | 35.3 | 27.4 | 57.1 | 47.3 | 65.9 | 24.7 | 40.2 |
| Unfrozen | 250M | 48.9 | **35.4** | **65.6** | 48.8 | 37.9 | **41.6** | 36.2 | 35.8 | 18.3 | 20.5 | 44.7 | 44.9 | 65.2 | 40.6 | 37.7 |
| Unfrozen | 500M | 48.1 | 33.9 | 64.1 | **50.0** | 39.9 | 40.7 | 38.2 | **39.8** | 21.9 | **22.4** | 47.5 | **45.0** | 65.1 | **36.5** | 39.2 |
| Unfrozen | 750M | 49.4 | 32.3 | 62.3 | 47.0 | **41.0** | 39.0 | **39.2** | 37.9 | **25.0** | **22.4** | **49.0** | **45.0** | 65.0 | 37.1 | 39.1 |
| Frozen | 100M | 50.7 | – | 62.2 | 47.4 | 29.9 | 37.8 | 32.8 | 25.1 | 12.2 | 15.6 | 38.6 | 44.7 | **67.0** | 36.9 | 39.7 |
| Frozen | 250M | **51.0** | – | 63.0 | 47.1 | 31.4 | 38.6 | 31.1 | 21.8 | 18.3 | 16.4 | 40.8 | 44.8 | 66.9 | 39.4 | 40.1 |
| Frozen | 500M | 49.8 | – | 61.6 | 45.9 | 32.1 | 37.5 | 27.4 | 18.7 | 17.1 | 16.4 | 39.9 | 44.7 | 66.8 | 38.8 | 40.2 |
| Frozen | 750M | 50.8 | – | 60.8 | 45.1 | 34.5 | 38.3 | 24.6 | 17.2 | 18.3 | 15.0 | 38.5 | 44.9 | 66.4 | 39.4 | **41.4** |

degradation on the affected generation benchmarks as Stage 3b training is extended (Table 15). Moreover, the unfrozen baselines generally retain reasoning performance better, suggesting that MLP adaptation is helpful rather than the root cause of the decline. Knowledge-heavy benchmarks exhibit mixed sensitivity to freezing: some remain competitive under frozen MLPs, indicating that mixer and routing updates can absorb part of the distillation signal, while others degrade, consistent with the need for MLP plasticity for broader knowledge transfer (especially cross-lingual). Finally, perplexity-based metrics change little across the duration sweep, again highlighting the gap between PPL-based and generation-based evaluation. These results argue against MLP knowledge corruption as the primary driver of Stage 3b degradation and support keeping MLPs trainable during Stage 3b.

## J   Stage 3a Budget Ablation

This section details the Stage 3a pretraining-distillation budget summarized in Section 5.2. The Stage 3b training-duration pattern is shown alongside the extended Stage 3b results in Appendix K, and the dataset ablation (Table 3) is in the main text.

We vary the Stage 3a token budget while holding the instruction-tuning stage (Stage 3b) fixed, so the student always learns the chat format and basic instruction-following behavior.[11]  Increasing the budget monotonically reduces held-out KL divergence (Figure 6), indicating better distributional alignment with the teacher. This alignment does not translate proportionally into stronger downstream generation after Stage 3b (Table 16). PIQA and MMLU are essentially unchanged across Stage 3a budgets, whereas ARC (Easy/Challenge) and WinoGrande improve more noticeably. Several perplexity-based benchmarks appear already close to the teacher after a short Stage 3a (e.g. PIQA), but LAMBADA reveals a substantial remaining perplexity gap that narrows only gradually with more Stage 3a training.

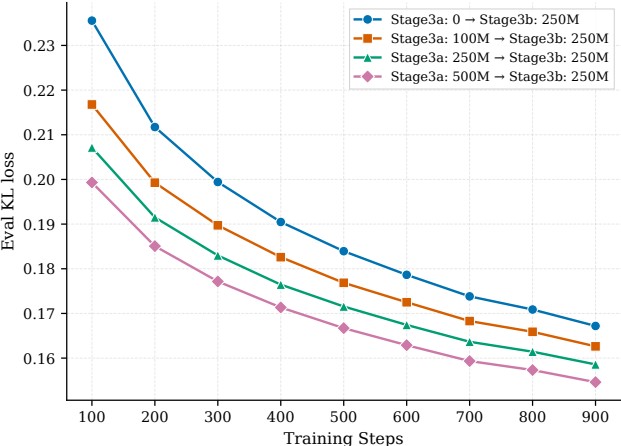

Figure 6: Held-out KL loss during Stage 3b for different Stage 3a token budgets.

Table 16: Stage 3a dataset size ablation: downstream results after Stage 3a (top, perplexity-based) and after Stage 3b (bottom, generation-based). We report accuracy for all tasks except LAMBADA, where we report perplexity ($\downarrow$).

| Checkpoint | | WG ↑ | HS ↑ | LAMBADA ↓ | ARC-E ↑ | ARC-C ↑ | PIQA ↑ | MMLU ↑ |
| Stage 3a | Stage 3b | | | | | | | |
|---|---|---|---|---|---|---|---|---|
| *Teacher (Qwen3-0.6B)* | | 55.5 | 47.3 | 24.7 | 55.6 | 34.2 | 65.9 | 40.2 |
| *After Stage 3a (LM Evaluation Harness, perplexity-based)* | | | | | | | | |
| 100M | – | 54.3 | 44.3 | 38.8 | 53.8 | 32.0 | 65.9 | 38.3 |
| 250M | – | 54.9 | 44.7 | 32.8 | 50.2 | 31.5 | 66.4 | 30.2 |
| 500M | – | 54.7 | 45.0 | 31.7 | 54.9 | 32.3 | 65.9 | 36.0 |
| *After Stage 3b (EvalScope, generation-based)* | | | | | | | | |
| 100M | 250M | 31.1 | 18.9 | 43.6 | 46.6 | 35.3 | 65.2 | 30.6 |
| 250M | 250M | 30.3 | 17.8 | 42.8 | 51.1 | 38.1 | 65.1 | 30 |
| 500M | 250M | 36.1 | 18.6 | 41.3 | 52.4 | 38.1 | 64.7 | 30.2 |

## K   Extended Stage 3b Results

Table 17 presents the complete results for all Stage 3b configurations across 11 generation-based and 7 perplexity-based benchmarks, the full source for the generation-only summary visualised in Figure 7 and

---

[11]These experiments use an earlier configuration with 5 attention layers. We later adopted 7 attention layers to match the 1:3 attention-to-SSM ratio used in other hybrid models (Team et al., 2025). Since this ablation compares relative rankings of configurations rather than absolute scores, the conclusions transfer to the 7-layer setup.

the protocol comparison in Figure 3. KD with completion-only masking achieves the best student result on every generation benchmark. For generation columns, greedy decoding results are shown in parentheses. We report them to confirm that the recipe ranking is not an artefact of sampling variance.

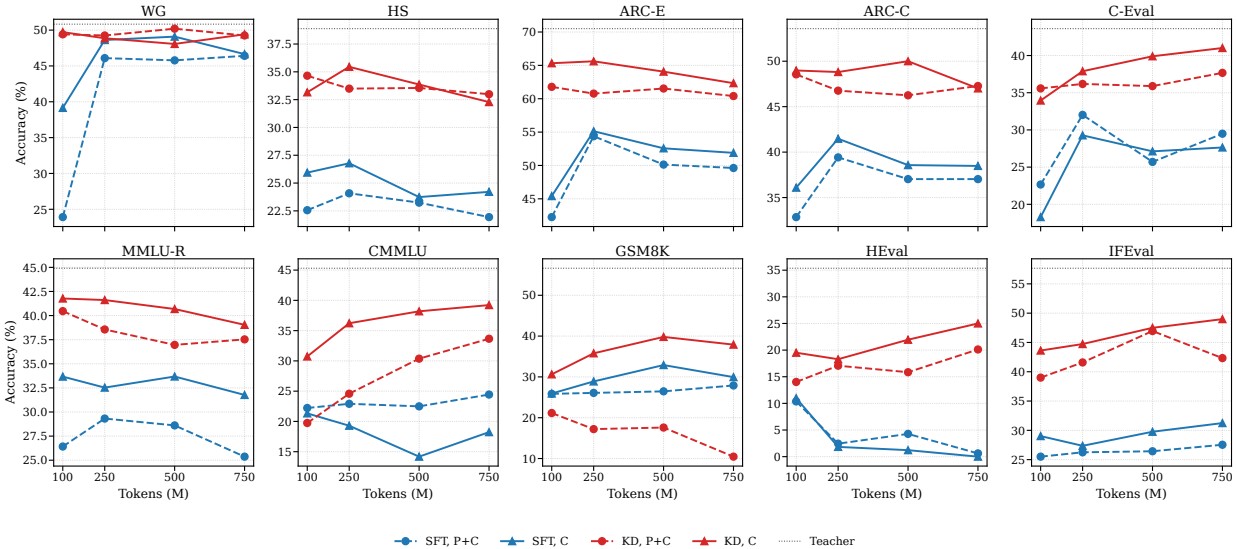

Figure 7: Stage 3b ablation, generation-based evaluation. Each panel shows accuracy on one of ten generation benchmarks as a function of the Stage 3b token budget. Colour encodes the training objective (KD red, SFT blue) and line style encodes the loss-masking strategy (P+C dashed, C solid). The dotted horizontal line is the Qwen3-0.6B teacher reference. Full numbers, perplexity-based scores, BBH, and greedy-decoding variants are in Table 17.

Table 17: Extended Stage 3b ablation. Generation-based results (EvalScope, left) with greedy decoding in ₚₐᵣₑₙₜₕₑₛₑₛ; perplexity-based results (LM-eval, right). Best student result per column in **bold**.

| | | | Generation-based (EvalScope) | | | | | | | | | | Perplexity-based (LM-eval) | | | | | | |
|---|---|---|---|---|---|---|---|---|---|---|---|---|---|---|---|---|---|---|---|
| Loss | Mask | Tokens | C-Eval | MMLU-R | CMMLU | WG | HS | ARC-E | ARC-C | BBH | GSM8K | HEval | IFEval | WG | HS | ARC-E | ARC-C | PIQA | LAM.↓ | MMLU |
| *Teacher* | | | 42.9 (43.5) | 46.1 (46.0) | 45.2 (46.2) | 51.2 (51.4) | 39.0 (39.3) | 70.8 (71.9) | 55.0 (56.2) | 27.4 (27.4) | 57.5 (59.7) | 35.3 (33.5) | 57.1 (58.2) | 55.5 | 47.3 | 55.6 | 34.2 | 65.9 | 24.7 | 40.2 |
| SFT | P+C | 250M | 32.0 (33.1) | 29.3 (34.6) | 22.9 (24.3) | 46.1 (48.2) | 24.1 (28.2) | 54.4 (60.5) | 39.4 (46.2) | 5.1 (5.8) | 26.1 (27.9) | 2.4 (0.0) | 26.2 (24.8) | 55.9 | 47.3 | 54.4 | 32.4 | 65.6 | 21.2 | 40.5 |
| SFT | P+C | 500M | 25.7 (29.0) | 28.6 (33.2) | 22.5 (23.4) | 45.8 (48.6) | 23.2 (28.6) | 50.1 (59.4) | 37.0 (45.1) | 8.2 (5.6) | 26.5 (26.8) | 4.3 (0.0) | 26.4 (24.9) | 56.0 | 47.7 | 52.5 | **34.0** | 65.3 | 20.8 | 39.8 |
| SFT | P+C | 750M | 29.5 (33.7) | 25.4 (27.3) | 24.4 (24.4) | 46.4 (48.5) | 21.9 (23.1) | 49.6 (55.6) | 37.0 (40.6) | 9.3 (8.8) | 27.9 (29.6) | 0.6 (0.0) | 27.5 (24.4) | 56.4 | **48.0** | 51.5 | 32.4 | 65.6 | **18.6** | 41.6 |
| KD | P+C | 250M | 36.2 (37.4) | 38.6 (38.6) | 24.6 (22.4) | 49.2 (49.9) | 33.5 (33.1) | 60.8 (61.5) | 46.8 (46.7) | 17.6 (18.2) | 17.2 (20.0) | 17.1 (16.5) | 41.6 (36.2) | **57.0** | 45.3 | **55.4** | 32.4 | 64.4 | 32.0 | 40.9 |
| KD | P+C | 500M | 35.9 (37.5) | 37.0 (38.2) | 30.4 (29.8) | **50.2 (50.0)** | 33.6 (33.6) | 61.5 (62.1) | 46.2 (47.9) | 17.5 (18.4) | 17.6 (19.6) | 15.8 (18.9) | 46.9 (39.0) | 56.3 | 45.2 | 54.5 | 31.7 | 64.4 | 30.0 | **42.4** |
| KD | P+C | 750M | 37.7 (38.0) | 37.5 (38.7) | 33.7 (33.7) | 49.2 (49.7) | 33.0 (33.0) | 60.4 (61.7) | 47.3 (47.1) | 18.4 (19.4) | 10.5 (13.0) | 20.1 (20.7) | 42.3 (38.1) | 56.4 | 45.3 | 54.0 | 32.6 | 64.5 | 28.9 | 41.6 |
| SFT | C | 250M | 29.3 (28.7) | 32.5 (36.0) | 19.3 (6.6) | 48.6 (49.5) | 26.8 (28.7) | 55.1 (60.7) | 41.5 (46.0) | 6.7 (5.3) | 28.9 (32.3) | 1.8 (0.6) | 27.4 (24.0) | 55.7 | 45.7 | 55.1 | 30.8 | 64.6 | 40.5 | 34.8 |
| SFT | C | 500M | 27.1 (29.7) | 33.7 (35.2) | 14.2 (2.9) | 49.1 (49.2) | 23.7 (24.4) | 52.6 (58.7) | 38.6 (42.8) | 11.4 (9.3) | 32.9 (35.3) | 1.2 (1.2) | 29.8 (25.1) | 55.2 | 46.1 | 54.6 | 31.6 | 64.9 | 43.5 | 34.1 |
| SFT | C | 750M | 27.6 (30.1) | 31.8 (34.6) | 18.2 (11.2) | 46.7 (48.0) | 24.2 (27.4) | 51.9 (57.1) | 38.5 (42.5) | 9.7 (7.6) | 29.9 (32.2) | 0.0 (0.0) | 31.2 (22.9) | 55.3 | 46.1 | 54.3 | 30.2 | 65.6 | 37.5 | 30.3 |
| KD | C | 250M | 37.9 (39.9) | **41.6 (42.1)** | 36.2 (36.0) | 48.9 (48.6) | **35.4 (36.6)** | **65.6 (67.6)** | 48.8 (50.2) | 20.5 (20.8) | 35.8 (37.8) | 18.3 (24.4) | 44.7 (41.6) | 56.4 | 44.9 | 54.9 | 31.7 | 65.2 | 40.6 | 37.7 |
| KD | C | 500M | 39.9 (40.9) | 40.7 (41.3) | 38.2 (39.0) | 48.1 (49.4) | 33.9 (35.0) | 64.1 (65.8) | **50.0 (50.6)** | **22.4 (22.5)** | **39.8 (40.3)** | 21.9 (22.6) | 47.5 (42.9) | 55.8 | 45.0 | 54.8 | 31.1 | 65.1 | 36.5 | 39.2 |
| KD | C | 750M | **41.0 (41.0)** | 39.0 (40.9) | **39.2 (39.7)** | 49.4 (49.6) | 32.3 (33.4) | 62.3 (63.4) | 47.0 (49.7) | **22.4 (22.9)** | 37.9 (39.4) | **25.0 (21.9)** | **49.0 (44.0)** | 56.3 | 45.0 | 54.9 | 32.3 | 65.0 | 37.1 | 39.1 |

## L   Combined Stage Mixing

The four-stage sequential pipeline (Section 3.2) trains on pretraining and instruction data in separate stages, which risks Stage 3b overwriting the general language representations acquired during Stage 3a, a pattern visible in the benchmark-specific degradation reported in Section 5.1. We investigate whether merging Stages 3a and 3b into a single training phase with interleaved pretraining and instruction data can circumvent this overwrite pattern while simplifying the pipeline. Table 18 compares the sequential pipeline against ten combined-stage configurations varying in total token budget, pretrain-to-instruct ratio, and pretraining data composition. Several patterns emerge. First, at matched compute budgets, the sequential pipeline outperforms combined mixing, with the largest gaps on reasoning and code generation benchmarks; the dedicated pretraining phase in Stage 3a appears to provide a stronger representational foundation than interleaved pretraining can replicate at the same scale. Second, among combined configurations, instruction-heavy ratios consistently outperform pretrain-heavy ones: allocating roughly 85% of tokens to instruction data and retaining only a small pretraining fraction yields the best balance across knowledge, reasoning,

Table 18: Combined stage mixing results. Sequential baselines (top) use the standard pipeline: 500M tokens of Stage 3a pretraining KD followed by Stage 3b instruction KD with completion-only masking and frozen attention. Combined stage runs (bottom) interleave pretraining and instruction data in a single training stage after Stages 1–2. Ratios denote pretrain:instruct data proportions; ⋆ indicates balanced FineWeb (50:50 EN:CN rather than the natural 29:71 ratio). Generation-based evaluation (EvalScope). Best student result per column in **bold**.

| Configuration | Tokens | C-Eval ↑ | MMLU-R ↑ | CMMLU ↑ | ARC-E ↑ | ARC-C ↑ | HS ↑ | WG ↑ | GSM8K ↑ | HEval ↑ | IFEval ↑ | LAM. ↓ |
|---|---|---|---|---|---|---|---|---|---|---|---|---|
| *Teacher (Qwen3-0.6B)* | | 42.9 | 46.1 | 45.2 | 70.8 | 55.0 | 39.0 | 51.2 | 57.5 | 35.3 | 57.1 | 24.7 |
| *Sequential pipeline (Stage 3a → Stage 3b)* | | | | | | | | | | | | |
| KD + C, 250M 3b | 750M | 37.9 | 41.6 | 36.2 | 65.6 | 48.8 | 35.4 | 48.9 | 35.8 | 18.3 | 44.7 | 40.63 |
| KD + C, 500M 3b | 1B | 39.9 | 40.7 | 38.2 | 64.1 | **50.0** | 33.9 | 48.1 | **39.8** | 21.9 | 47.5 | 36.46 |
| KD + C, 750M 3b | 1.25B | **41.0** | 39.0 | **39.2** | 62.3 | 47.0 | 32.3 | 49.4 | 37.9 | **25.0** | **49.0** | 37.11 |
| *Combined stage mixing (pretrain:instruct ratio)* | | | | | | | | | | | | |
| 50/50 | 500M | 36.0 | 40.4 | 28.9 | 65.2 | 46.9 | 34.1 | 49.2 | 26.2 | 17.7 | 42.9 | 32.65 |
| 67/33 | 750M | 38.1 | **41.9** | 31.1 | 65.2 | 48.6 | 31.5 | **51.8** | 27.9 | 17.7 | 42.1 | 36.66 |
| 50/50 | 750M | 40.1 | 39.6 | 34.5 | 64.9 | 48.3 | 29.5 | 51.0 | 27.7 | 17.1 | 41.4 | 32.75 |
| 50/50⋆ | 750M | 38.8 | 39.4 | 36.9 | 64.8 | 49.6 | 32.7 | 50.4 | 29.1 | 14.6 | 44.7 | 33.12 |
| 33/67 | 750M | 37.5 | 39.5 | 35.4 | 65.2 | 49.3 | 31.8 | 50.4 | 30.9 | 13.4 | 44.5 | 34.81 |
| 50/50 | 1B | 37.1 | 41.2 | 32.8 | 65.4 | 49.1 | 35.3 | 48.8 | 28.4 | 17.7 | 46.2 | 32.20 |
| 25/75⋆ | 1B | 39.0 | 40.8 | 38.4 | 65.7 | 49.7 | 30.2 | 49.4 | 32.2 | 16.5 | 45.7 | 34.52 |
| 15/85⋆ | 1B | 37.5 | 41.7 | 38.3 | 65.3 | 49.7 | 31.5 | 50.8 | 34.0 | 15.8 | 48.2 | 32.86 |
| 10/90⋆ | 1B | 37.4 | 40.8 | 36.6 | 66.0 | 49.9 | 32.8 | 47.4 | 35.1 | 16.5 | 45.5 | 34.45 |
| 15/85⋆ | 2B | 38.2 | 41.2 | **39.2** | **67.2** | **50.0** | **36.1** | 50.0 | 37.1 | 18.3 | 48.2 | **31.13** |

and instruction-following tasks. Third, balancing the English-to-Chinese ratio in the pretraining subset (50:50 rather than the natural distribution) consistently improves Chinese knowledge benchmarks without harming English performance, confirming that data composition within the pretraining fraction matters even when that fraction is small. Fourth, unlike the sequential pipeline where extending Stage 3b degrades several benchmarks (Table 17), combined mixing scales gracefully with additional compute, showing steady improvement from medium to large token budgets without the characteristic degradation pattern. Fifth, at sufficient total compute, the best combined configuration approaches or matches the sequential pipeline's best results across most benchmarks and surpasses it on language modeling and several reasoning tasks, suggesting that combined mixing may become the preferred strategy in higher-compute regimes where the benefits of graceful scaling outweigh the sequential pipeline's advantage at smaller budgets.

## M   Stage 3b Recipe Ablations at 1.7B

To test whether the Stage 3b recipe conclusions extend beyond the 0.6B scale used for the systematic study, we repeat the three highest-leverage ablations with a Qwen3-1.7B teacher. We distil into the same Hybrid-KDA architecture and keep every training setting other than the ablated axis fixed. Stage 3b uses 500M tokens, as at 0.6B, and all numbers are averaged over two seeds. Table 19 reports the training-objective and loss-masking comparisons, and Table 20 reports the attention-freezing comparison.

Table 19: Stage 3b recipe ablation at 1.7B (training objective and loss masking), Hybrid-KDA student distilled from a Qwen3-1.7B teacher, 500M Stage 3b tokens, frozen attention, seed-averaged. Generation-based results (EvalScope, left) and perplexity-based results (LM-eval, right). Masking: P+C = full-sequence (prompt + completion); C = completion-only. Best student result per column in **bold**.

| Loss | Mask | Tokens | Generation-based (EvalScope) | | | | | | | | | | | Perplexity-based (LM-eval) | | | | | | |
|---|---|---|---|---|---|---|---|---|---|---|---|---|---|---|---|---|---|---|---|---|
| | | | C-Eval | MMLU-R | CMMLU | WG | HS | ARC-E | ARC-C | BBH | GSM8K | HEval | IFEval | WG | HS | ARC-E | ARC-C | PIQA | LAM.↓ | MMLU |
| *Teacher (Qwen3-1.7B)* | | | 62.8 | 66.9 | 62.0 | 54.2 | 59.8 | 84.8 | 73.6 | 34.3 | 80.9 | 58.5 | 66.1 | 61.6 | 60.5 | 69.7 | 42.8 | 72.6 | 12.2 | 55.4 |
| SFT | C | 500M | 33.1 | 45.7 | 46.1 | 49.5 | 31.4 | 74.8 | 60.6 | 9.5 | 54.0 | 0.0 | 36.2 | **61.2** | **60.3** | 63.9 | 38.7 | 71.4 | **11.6** | 51.5 |
| KD | C | 500M | **56.5** | **55.0** | 54.0 | 33.2 | 23.5 | 68.5 | 58.2 | **21.2** | **60.1** | **34.8** | **58.6** | 60.9 | 58.2 | **68.5** | 41.3 | 71.6 | 13.4 | 53.0 |
| KD | P+C | 500M | **56.5** | 52.2 | 54.8 | 51.2 | **32.3** | **80.5** | **68.1** | 19.6 | 58.0 | 7.3 | 56.4 | 60.7 | 58.9 | 68.0 | **41.7** | **72.1** | 12.9 | **53.4** |

The objective conclusion reproduces. KD leads SFT on generation by 7.5pp on average, with the largest gaps on instruction following, code, and Chinese knowledge, while SFT attains the lower perplexity on the language-modeling benchmarks (LAMBADA, HellaSwag, WinoGrande). The protocol disagreement that motivates the paper is therefore present at 1.7B in the same direction as at 0.6B.

The masking conclusion reproduces only in part. Completion-only retains a clear advantage on instruction following, code, and reasoning, but full-sequence masking overtakes it on the commonsense multiple-choice benchmarks (WinoGrande, HellaSwag, ARC) and so attains a slightly higher generation average. On those

benchmarks the completion-only student generates answers at close to random accuracy, near 33 on the binary WinoGrande and near 23 on the four-way HellaSwag, while scoring normally under perplexity-based evaluation. This pattern matches the multiple-choice extraction failures characterised in Appendix N rather than a loss of commonsense ability, and it makes completion-only masking the most evaluation-sensitive of the three recipe choices at scale.

Table 20: Effect of freezing attention layers during Stage 3b at 1.7B. Both runs use KD with completion-only masking and 500M Stage 3b tokens, differing only in whether the attention layers are frozen, seed-averaged. Benchmarks match Table 19: generation-based (EvalScope, left) and perplexity-based (LM-eval, right). Best result per column in **bold**.

| Frozen? | Generation-based (EvalScope) | | | | | | | | | | | Perplexity-based (LM-eval) | | | | | | |
|---|---|---|---|---|---|---|---|---|---|---|---|---|---|---|---|---|---|---|
| | C-Eval | MMLU-R | CMMLU | WG | HS | ARC-E | ARC-C | BBH | GSM8K | HEval | IFEval | WG | HS | ARC-E | ARC-C | PIQA | LAM.↓ | MMLU |
| No | **56.8** | 54.9 | **55.0** | 32.1 | 18.0 | 63.7 | 53.2 | **21.9** | 58.7 | **35.1** | **58.7** | **61.2** | 57.9 | **69.0** | **42.2** | 71.5 | **12.9** | **53.3** |
| Yes | 56.5 | **55.0** | 54.0 | **33.2** | **23.5** | **68.5** | **58.2** | 21.2 | **60.1** | 34.8 | 58.6 | 60.9 | **58.2** | 68.5 | 41.3 | **71.6** | 13.4 | 53.0 |

The freezing conclusion reproduces. The gains are large on the commonsense multiple-choice benchmarks (HellaSwag and ARC, around 5pp each in Table 20). On the remaining benchmarks the two runs lie within their seed-to-seed variation, and the few on which the unfrozen run scores higher do so by margins smaller than that variation, so freezing never significantly underperforms. Averaged over all eleven generation benchmarks freezing improves by 1.4pp, rising to 2.7pp over the knowledge and commonsense subset (C-Eval, MMLU-R, WinoGrande, HellaSwag, ARC). Perplexity-based scores move by less than half a point and, if anything, slightly favour the unfrozen run. This is the same asymmetry reported at 0.6B in Table 2, where freezing improves generation while leaving perplexity-based evaluation almost unchanged.

## N    Qualitative KD vs. SFT Failure Patterns

The Stage-3b ablation in Section 5.1 reports a consistent advantage for knowledge distillation over supervised fine-tuning on generation-based evaluation, with the opposite pattern on several perplexity-based benchmarks. To complement the aggregate numbers, we inspected the outputs of two same-recipe Hybrid-KDA 0.6B Stage-3b checkpoints (completion-only, 500M tokens) that differ only in the loss type, on two unrelated tasks (HellaSwag and GSM8K). Both tasks show an identical +7.2 pp accuracy gap in KD's favour ($N = 1000$ and $N = 1319$ respectively, greedy bf16 generation, seed = 1). The output traces reveal four behaviours that recur in the SFT generations but are essentially absent from the KD generations on the same prompts. Table 21 summarises the rates.

**Template overfitting.** SFT wraps 100% of its outputs in literal `<answer>...</answer>` tags on both tasks, including on GSM8K where the 4-shot prompt demonstrates the plain "Reasoning: ... ANSWER: \boxed{N}" format from `lm-evaluation-harness`. KD wraps none. The tags appear in part of the upstream Stage-3b instruct corpus (AM-Qwen3-Distilled); SFT has learned them as a literal output prefix and suffix, whereas KD trains against the teacher's full output distribution rather than the labels themselves and the Qwen3 teacher does not emit the tag.

**Multiple-choice positional collapse.** On HellaSwag, SFT picks option "A" on 79.5% of all questions (against KD's 54%), and on 86% of gold-A questions specifically. Conditional accuracy on the non-A buckets collapses to 5.1% on gold-D and 9.1% on gold-B for SFT, while KD retains discrimination across the four options.

**Wordier and less accurate chains of thought.** On GSM8K, SFT generates a mean of 404.3 output tokens against KD's 302.5, with twice the 5-gram local repetition rate and 4.7× the rate of outputs that fail to emit \boxed{·}. Inspection of per-prompt disagreement cases (KD-correct, SFT-wrong) shows the SFT chains restating the problem with extra ceremony, introducing ill-defined intermediate variables, and committing arithmetic errors that KD avoids.

**Few-shot recency mimicry.** On the 4-shot GSM8K prompt, SFT regurgitates tokens lifted directly from the recency-most demonstration into answers to unrelated questions roughly three times as often as

KD (strict-trigger rate 3.0% vs. 1.0%), leading to incorrect answers. Re-running the same benchmark with no in-context demonstrations (`few_shot_num=0`) drops the mimicry rate to zero on both models and shrinks the KD-vs-SFT accuracy gap from +7.2 to +3.4 pp, confirming that the demonstrations themselves are the trigger and that SFT is the more susceptible of the two models to anchoring on irrelevant phrasing in the prompt.

|  | KD | SFT |
|---|---|---|
| *HellaSwag multiple-choice extraction* | | |
| Picks "A" overall | 54% | 79.5% |
| Accuracy on gold = A | 69.6% | 85.6% |
| Accuracy on gold ≠ A | 28.4% | 12.9% |
| *GSM8K output statistics (4-shot)* | | |
| Mean output tokens | 302.5 | 404.3 |
| 5-gram repetition rate (mean) | 0.027 | 0.054 |
| Contains \boxed{·} | 98.2% | 91.6% |
| *Recency mimicry of in-context demonstrations on GSM8K* | | |
| Strict-trigger rate, 4-shot | 1.0% | 3.0% |
| Strict-trigger rate, 0-shot | 0.0% | 0.0% |

Table 21: Rates of the four SFT-specific behaviours described in this appendix. Both checkpoints are Hybrid-KDA 0.6B Stage-3b (completion-only, 500M tokens), identical generation config, differing only in the Stage-3b loss type.

These four behaviours are consistent with a single hypothesised mechanism: KL distillation transmits properties of the teacher's full output distribution to the student, including its calibration over option tokens on multiple choice (Zheng et al., 2024) and its resistance to copying labels from in-context demonstrations (Ali et al., 2024), whereas supervised fine-tuning against arg-max labels discards those properties. Instruction-tuned models additionally remain sensitive to the surface formatting of few-shot prompts (Sclar et al., 2023), and SFT is documented to emit training templates verbatim when the instruction-template diversity of the training set is low (Lin et al., 2025a), consistent with the `<answer>...</answer>` wrap we observe. The wordier-but-less-accurate chain pattern on GSM8K is reminiscent of the "flaw repetition" failure mode Yao et al. (2025) document under SFT-only post-training of reasoning models, alongside their finding that SFT-only pipelines degrade calibration relative to the base distribution (their analysis is at 7B–70B reasoning-model scale with reasoning-specific recipes, so the parallel is qualitative).

**Representative example traces**

The four behaviours described above are illustrated below on four concrete prompts. Each block shows the question (4-shot in-context demonstrations stripped for readability), the KD checkpoint's full output, and the SFT checkpoint's output verbatim. Outputs are shown in monospace to preserve the literal characters emitted by the model, including the spurious `<answer>...</answer>` wrap.

**Example 1: Template wrap (GSM8K, gold = 8).** *Q:* Brandon's iPhone is four times as old as Ben's iPhone. Ben's iPhone is two times older than Suzy's iPhone. If Suzy's iPhone is 1 year old, how old is Brandon's iPhone?

KD output:

```
- Suzy's age: 1 year.
- Ben's age: 2 x 1 = 2 years.
- Brandon's age: 4 x 2 = 8 years.
\boxed{8}
```

SFT output:

```
<answer>Brandon's iPhone is four times as old as Ben's iPhone.
Let x = age of Ben's iPhone, so Brandon's iPhone is 4x years old.
Suzy's iPhone is 1 year old.
Since Suzy's iPhone is two times older than Ben's iPhone:
    2 * x = 1   =>   x = 1/2
Thus Ben's iPhone is 1/2 years old, and Brandon's is 4 * 1/2 = 2.
The final answer is \boxed{2}.</answer>
```

SFT wraps every output in literal `<answer>...</answer>` tags and additionally inverts the "two times older than" relation. KD never produces the wrap and computes the relation correctly.

**Example 2: Multiple-choice positional collapse (HellaSwag, gold = D, activity "Clean and jerk").** *Context:* A lady walks to a barbell. She bends down and grabs the pole. The lady. . .

*Candidate continuations:*

- **A.** swings and lands in her arms.
- **B.** pulls the barbell forward.
- **C.** pulls a rope attached to the barbell.
- **D.** stands and lifts the weight over her head.

**KD answer:** D (correct). **SFT answer:** A.

Even though option A is semantically incongruous with a clean-and-jerk lift, SFT collapses to A; KD with the same base recipe correctly identifies D.

**Example 3: Wordier but less accurate chain of thought (GSM8K, gold = 50).** *Q:* A car is on a road trip and drives 60 mph for 2 hours, and then 30 mph for 1 hour. What is the car's average speed in mph during this trip?

KD output:

```
1. Distance in first 2 hours: 60 x 2 = 120 miles
2. Distance in next 1 hour:   30 x 1 = 30 miles
3. Total distance: 120 + 30 = 150 miles
4. Total time:     2 + 1 = 3 hours
5. Average speed:  150 / 3 = 50 mph
\boxed{50}
```

SFT output:

```
<answer>To determine the car's average speed during the trip ...
### Step 1: Understand the Given Information
### Step 2: Calculate the Distance Traveled in Each Segment
  First segment:  60 mph * 2 h = 120 miles
  Second segment: 30 mph * 1 h = 30  miles
### Step 3: Total Distance Traveled
  120 + 30 = 150 miles
### Step 4: Calculate the Average Speed
  Average Speed = 150 miles / 2 hours = 75 mph
### Final Answer
\boxed{75}</answer>
```

SFT produces a longer, more heavily formatted answer but divides total distance by the duration of one segment (2 h) rather than total trip time (3 h). Surface structure is not understanding.

**Example 4: Few-shot recency mimicry (GSM8K, gold = 32).**   *Q:* Anakin and Locsin went to the beach today. Anakin caught 10 starfish, 6 sea horses, and 3 clownfish. Locsin caught 5 fewer starfish than Anakin, 3 fewer sea horses than Anakin, and 2 more clownfish than Anakin. How many fish were they able to catch in total?

KD output:

```
Anakin: 10 + 6 + 3 = 19 fish.
Locsin: (10-5) + (6-3) + (3+2) = 5 + 3 + 5 = 13 fish.
Total: 19 + 13 = 32.   \boxed{32}
```

SFT output (trimmed; the relevant intrusion is at steps 5–6):

```
<answer>**Step-by-Step Explanation:**
1. Anakin's Catch: 10 starfish, 6 sea horses, 3 clownfish.
2. Locsin's Catch: 10-5=5 starfish, 6-3=3 sea horses, 3+2=5 clownfish.
3. Total Catch by Anakin: 10 + 6 + 3 = 19 fish.
4. Total Catch by Locsin: 5 + 3 + 5 = 13 fish.
...
5. Remaining Pages for Tomorrow:
   - Total pages: 120
   - Pages already read: 19
   - Remaining pages: 120 - 19 = 101
   - Pages to read tomorrow: 101 / 2 = 50.5
6. Final Answer: 50.5 ~= 50 pages.
**Answer:** \boxed{50}</answer>
```

SFT correctly assembles the per-person fish counts in steps 1–4 and then silently switches into the "120-page book / half tomorrow" template lifted from the recency-most 4-shot demonstration (Julie reading a 120-page book), abandons the actual question, and reports 50. The intrusion is the prompt's fault rather than the training set's: at `few_shot_num=0` the mimicry rate falls to zero on both models (Table 21).

## O  Perplexity-Based Architecture Comparison

Table 4 in the main text reports generation-based results. Here we present the corresponding perplexity-based (LM Evaluation Harness) results for the same eight architectures.

Table 22: Architecture comparison under perplexity-based evaluation (LM Evaluation Harness). Same models as Table 4. We report accuracy (%) for all tasks except LAMBADA, where we report perplexity ($\downarrow$). Best student result per column in **bold**.

| Model | WG ↑ | HS ↑ | ARC-E ↑ | ARC-C ↑ | PIQA ↑ | LAM. ↓ | MMLU ↑ | CMMLU ↑ |
|---|---|---|---|---|---|---|---|---|
| *Teacher (Qwen3-0.6B)* | 55.5 | 47.3 | 55.6 | 34.2 | 65.9 | 24.7 | 40.2 | –[†] |
| Pure KDA | 55.1 | 42.9 | 49.5 | 29.4 | 64.3 | 53.7 | 35.9 | 33.3 |
| Pure Mamba | 53.8 | 39.2 | 39.8 | 26.4 | 63.3 | 135.0 | 25.0 | 26.4 |
| Hybrid Mamba | 54.0 | 42.9 | 53.6 | 30.5 | 65.7 | 73.0 | 34.7 | 38.8 |
| Hybrid KDA | **57.2** | **45.2** | 52.5 | **31.7** | **66.3** | **38.0** | **40.6** | **40.0** |
| Hybrid GDN | 53.3 | 41.6 | 51.2 | 28.7 | 64.0 | 56.1 | 30.0 | 34.9 |
| Hybrid GLA | 54.1 | 43.6 | **54.3** | 30.5 | 65.7 | 51.9 | 38.0 | 39.2 |
| Hybrid Lightning | 55.1 | 40.4 | 51.4 | 29.0 | 64.1 | 111.4 | 26.7 | 36.0 |

[†]Teacher CMMLU not evaluated under LM-Eval.

Architecture rankings under perplexity-based evaluation are broadly consistent with generation-based results: Hybrid KDA leads on most metrics and Pure Mamba is weakest. However, inter-model differences are markedly compressed: PPL-based scoring narrows the architecture gaps substantially compared to generation-based evaluation, with the exception of Pure Mamba's catastrophic LAMBADA perplexity. This

reinforces the paper's central finding that perplexity-based metrics underestimate quality differences between architectures.

## P  Maximum Throughput Under GPU Saturation

The batch-size-1 measurements in Section 5.3.1 do not capture the primary deployment benefit of reduced KV cache memory: fitting larger batches. Figure 8 reports peak throughput at each context length by sweeping batch sizes up to OOM.

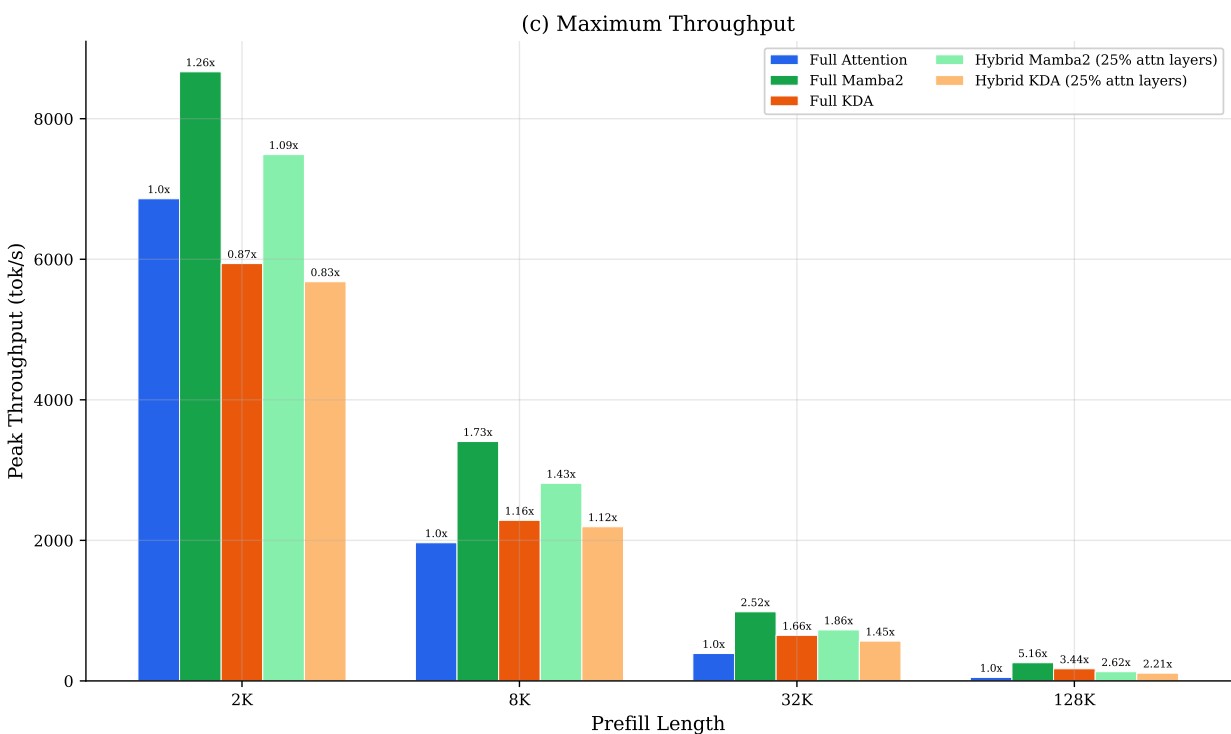

Figure 8: **(a)** Theoretical cache memory vs. sequence length. Attention cache grows linearly with context; SSM state is constant. **(b)** Peak throughput (sweeping batch size to OOM) at four context lengths. Annotations show relative throughput vs. the Teacher. At 32K+ tokens, SSM variants achieve 1.9–2.5× the teacher's peak throughput by fitting larger batches.

At short contexts (2K), KV cache memory is small relative to model weights, so all architectures achieve similar peak throughput. As context length grows, KV cache dominates memory allocation for the Transformer, limiting its maximum batch size. The SSM and hybrid variants, with ∼75% smaller cache, sustain larger batches and achieve up to 5× higher peak throughput at 128K tokens.

## Q  Long-Context Quality

The efficiency gains reported in Section 5.3.1 are meaningful only if hybrid models preserve functional quality at long contexts. We evaluate on LongBench (Bai et al., 2023) (real-world QA and summarization, 8K–22K tokens) and RULER (Hsieh et al., 2024) (synthetic needle-in-a-haystack retrieval, 4K–128K tokens). All students were distilled on sequences of at most 4K tokens with no dedicated long-context extension stage; the gap to the teacher reported below should be read in that light (see Section 6).

On LongBench (Table 23), all students fall substantially short of the teacher. The architecture hierarchy mirrors the short-context results (Table 4): Pure Mamba collapses (Avg 3.0), Pure KDA is moderate (12.6), and both hybrid variants lead, with Hybrid KDA (14.5) narrowly ahead of Hybrid Mamba (14.0). Individual

Table 23: LongBench results (F1 or ROUGE, ×100; contexts 8K–22K tokens). DuR = DuReader, HQA = HotpotQA, MuS = MuSiQue, NQA = NarrativeQA, QMS = QMSum, TQA = TriviaQA. "Hybrid" models retain 7 attention layers; "Pure" models use 0. Best student per column in **bold**.

| Model | DuR | HQA | MuS | NQA | QMS | TQA | Avg |
|---|---|---|---|---|---|---|---|
| *Teacher (Qwen3-0.6B)* | 32.8 | 24.4 | 8.6 | 13.0 | 21.0 | 59.6 | 26.6 |
| Pure Mamba | 4.4 | 0.6 | 0.6 | 0.9 | 9.7 | 1.9 | 3.0 |
| Pure KDA | 16.1 | 7.6 | 3.3 | 9.3 | **19.4** | 19.9 | 12.6 |
| Hybrid Mamba | **21.5** | 10.8 | 3.0 | 6.6 | 15.5 | **26.6** | 14.0 |
| Hybrid KDA | 21.4 | **12.5** | **5.2** | **10.3** | 18.6 | 19.1 | **14.5** |

task rankings vary (Hybrid Mamba leads on DuReader and TriviaQA, while Hybrid KDA is stronger on HotpotQA, MuSiQue, and NarrativeQA), but the overall gap to the teacher remains large, underscoring that short-context distillation alone does not fully transfer long-range comprehension.

Table 24: RULER NIAH single-needle retrieval accuracy (%) across context lengths. The three Single variants differ in haystack complexity and needle value type: **Single-1** embeds a numeric needle in a short repeated sentence, **Single-2** embeds a numeric needle in natural prose (Paul Graham essays), and **Single-3** embeds a UUID string in the same natural prose. "Hybrid" models retain 7 attention layers; "Pure" models use 0. Best student per column in **bold**.

| | NIAH-Single-1 | | | | | | NIAH-Single-2 | | | | | | NIAH-Single-3 | | | | | |
|---|---|---|---|---|---|---|---|---|---|---|---|---|---|---|---|---|---|---|
| Model | 4K | 8K | 16K | 32K | 64K | 128K | 4K | 8K | 16K | 32K | 64K | 128K | 4K | 8K | 16K | 32K | 64K | 128K |
| *Teacher* | 100.0 | 100.0 | 100.0 | 100.0 | 100.0 | 0.0[†] | 100.0 | 100.0 | 100.0 | 100.0 | 99.4 | 0.0[†] | 99.4 | 99.8 | 100.0 | 100.0 | 78.8 | 0.0[†] |
| Pure Mamba | 0.0 | 0.0 | 0.0 | 0.0 | 0.0 | 0.0 | 0.0 | 0.0 | 0.0 | 0.0 | 0.0 | 0.0 | 0.0 | 0.0 | 0.0 | 0.0 | 0.0 | 0.0 |
| Pure KDA | 96.4 | 88.8 | 60.2 | 29.0 | 11.2 | 5.4 | 0.6 | 0.0 | 0.0 | 0.0 | 0.0 | 0.0 | 0.0 | 0.0 | 0.0 | 0.0 | 0.0 | 0.0 |
| Hybrid Mamba | **100.0** | **100.0** | **100.0** | 97.6 | 75.8 | 14.0 | **100.0** | **100.0** | 95.8 | 37.6 | 4.2 | 1.0 | **86.8** | 74.6 | 25.2 | 14.8 | **3.2** | 0.0 |
| Hybrid KDA | 99.2 | **100.0** | 98.8 | 94.8 | **92.6** | **43.0** | 99.0 | 99.8 | **98.6** | **58.4** | **26.4** | **2.4** | 80.2 | **82.0** | **58.0** | **19.6** | **5.6** | 0.0 |

[†]Qwen3-0.6B supports up to 40K positions; 128K results are out-of-distribution.

The NIAH results (Table 24) provide a finer-grained view. Pure Mamba fails completely across all needle types and context lengths (0.0%), confirming that Mamba2 layers lack the precise token-level retrieval mechanism required for needle-in-a-haystack tasks. Pure KDA retains some retrieval ability on the simplest needle type (Single-1: 96.4% at 4K) but collapses to near-zero on Single-2 (0.6%) and Single-3 (0.0%). The dramatic difference is explained by the task structure: in Single-1, the haystack is a short sentence repeated many times, making any inserted needle highly salient against a trivially predictable background; in Single-2/3, the needle must be located within coherent natural prose, requiring precise positional attention that linear sequence mixers fundamentally lack. Single-3 adds further difficulty by requiring exact reproduction of 36-character UUID strings rather than short numbers, a task that demands character-level fidelity from the retrieval mechanism. This progression exposes a fundamental limitation of KDA layers: their success on Single-1 reflects pattern-matching against a predictable context rather than genuine retrieval capability. Both hybrid models perform dramatically better, with near-perfect accuracy on Single-1 and Single-2 at short contexts. Hybrid Mamba slightly outperforms Hybrid KDA at short contexts on the easiest tasks, but Hybrid KDA degrades more gracefully at longer contexts, maintaining 92.6% at 64K on Single-1 versus 75.8% for Hybrid Mamba. On the hardest needle type (Single-3), both hybrid models struggle beyond 16K tokens, with Hybrid KDA retaining a slight edge at 8K–32K.

Taken together, the long-context results confirm that hybrid architectures enable meaningful efficiency gains (Section 5.3.1) while retaining basic retrieval capability. Retained attention layers are essential for non-trivial retrieval, with both pure architectures failing on harder needle types. Among hybrid variants, KDA maintains a slight overall edge, particularly at longer contexts. A substantial gap to the teacher persists at harder retrieval patterns and longer contexts; we discuss in Section 6 why this is expected given the absence of a dedicated long-context training stage, and how concurrent work (Chen et al., 2026; Team et al., 2026) closes the gap with such a stage.

