# OpenReview forum: "When Perplexity Lies: Generation-Focused Distillation of Hybrid Sequence Models"
_TMLR — Under review for TMLR_

### Review · Reviewer_mgkZ · 2026-05-11

**Summary Of Contributions:**

### Summary of Contributions
The paper investigates cross-architecture distillation from Transformer LLMs into efficient hybrid recurrent/linear-attention models and argues that perplexity-based evaluation substantially misrepresents true generation quality in this setting. To address this, the authors propose a Hybrid-KDA architecture together with GenDistill, a multi-stage distillation pipeline that combines hidden-state alignment, end-to-end knowledge distillation, teacher-aligned instruction tuning, and selective retention of attention layers. Through extensive ablations across architectural and training design choices, the paper demonstrates that generation-based evaluation reveals large quality differences often hidden by log-likelihood-based evaluation, and identifies several key ingredients for successful distillation, including KD objectives, completion-only masking, teacher-aligned instruction data, and freezing the retained attention layers during post-training.

### Strengths
1) __Important and timely problem__. The paper addresses a practically important challenge: distilling Transformer LLMs into more efficient hybrid/recurrent architectures while preserving generation quality.
2) __Strong empirical study__. The paper includes extensive and well-controlled ablations across architecture, objectives, masking, datasets, training duration, and freezing strategies, providing a thorough analysis of the distillation pipeline.
3) __Compelling evaluation findings__. The central observation that perplexity-based evaluation can substantially misrepresent generation quality is convincing and consistently supported across experiments.
4) __Actionable practical insights__. The work identifies several clear design recommendations for effective distillation, including KD objectives, completion-only masking, teacher-aligned instruction data, and freezing retained attention layers.

### Weaknesses
1) __Inconsistency in evaluation philosophy__. Although the paper argues strongly against perplexity-based evaluation, some components of the pipeline (e.g., attention layer selection) still rely on perplexity as the optimization criterion.
2) __Limited architectural breadth__. The comparisons focus primarily on KDA and Mamba2, while several other recent efficient sequence modeling approaches are not evaluated.
3) __Limited scale of controlled experiments__. Most ablations are conducted only at the 0.6B scale, making it unclear whether the proposed recipe and conclusions fully transfer to larger models.
4) __Limited qualitative analysis__. The paper focuses heavily on benchmark metrics and provides little qualitative analysis of generation failures or behavioral differences between models.
5) __Long-context quality remains weak__. Despite strong efficiency gains, the distilled models still exhibit substantial degradation on long-context tasks, limiting their practical impact in some applications.

**Audience:**

Yes

**Audience Explanation:**

The paper addresses an important topic in efficient LLM design and distillation, namely, how to convert Transformer models into more efficient hybrid architectures while preserving generation quality. The findings regarding the mismatch between perplexity-based and generation-based evaluation are particularly relevant for researchers working on efficient language modeling, distillation, and evaluation methodology.

**Broader Impact Concerns:**

I do not have major broader-impact concerns beyond the standard considerations associated with large language models. The work primarily focuses on improving the efficiency and evaluation methodology for distilled language models rather than introducing new application domains or capabilities that entail obvious additional societal risks. A brief discussion acknowledging that more efficient LLM deployment may increase accessibility and deployment scale would be sufficient.

**Claims And Evidence:**

Yes

**Claims Explanation:**

The paper provides generally strong and convincing empirical evidence for its main claims through extensive controlled ablations across architectures, objectives, masking strategies, datasets, and freezing configurations. In particular, the central claim that perplexity-based evaluation can substantially misrepresent generation quality is consistently supported across experiments. The evidence is clearly presented and supported by both quality and efficiency evaluations. However, the conclusions are somewhat limited because most controlled experiments are conducted only at the 0.6B scale.

**Requested Changes:**

1) __Evaluate key findings at larger scales (Critical)__.
    Most controlled ablations are conducted only at the 0.6B scale. Since the paper’s main recommendations are presented broadly, the authors should provide at least partial validation that the observed trends (e.g., KD vs. SFT, completion-only masking, freezing attention layers) remain consistent at larger model sizes.
2) __Clarify the role of perplexity in the methodology (Important)__.
    The paper strongly argues against perplexity-based evaluation, yet some parts of the pipeline (particularly attention layer selection) still optimize for perplexity. The paper would benefit from a clearer discussion of why perplexity remains appropriate in these intermediate optimization steps despite the broader critique.
3) __Expand architectural comparisons (Non-critical)__.
    The architectural evaluation focuses primarily on KDA and Mamba2. Including additional modern recurrent or linear-attention baselines, or discussing why they were omitted, would strengthen the work.
4) __Add qualitative analysis of generation behavior (Non-critical)__.
    The paper would benefit from a brief qualitative section that shows representative generation failures or examples in which perplexity-based evaluation yields misleading conclusions compared to autoregressive generation.
5) __Further analyze long-context degradation (Non-critical)__.
    Although the efficiency gains are compelling, long-context quality remains substantially below that of the teacher model. Additional discussion or analysis of these remaining limitations would improve the paper.

---

> ### Author Response · Authors · 2026-05-27
> **Response to reviewer mgKZ**
>
> We thank the reviewer for the careful and constructive review, and for recognising the problem as timely, the empirical study as thorough, and the perplexity-versus-generation finding as convincing. We have run new experiments and revised the manuscript in response to every point. The summary below describes the changes and reports the new results.
>
>
> ### Evaluate key findings at larger scale (Critical)
>
> We agree that a recipe study at a single scale leaves open whether the conclusions are scale-specific, so we ran a scale-up experiment at the 1.7B scale. We repeated three of the Stage-3b ablations (training objective, attention freezing, and loss masking), holding every other training setting fixed and averaging over two seeds at a matched 500M-token Stage-3b budget. All three axes are reported in Table 1 below, each compared against the shared KD, C frozen-attention configuration.
>
> **Table 1. Stage-3b recipe ablation at 1.7B (training objective, loss masking, and attention freezing).** Hybrid-KDA student distilled from a Qwen3-1.7B teacher, 500M Stage-3b tokens, seed-averaged. The first three student rows hold attention frozen and vary objective and masking; the last row repeats the KD, C configuration with attention unfrozen. Mask: P+C = full-sequence, C = completion-only. Best student per column in **bold**.
>
> _Generation-based (EvalScope):_
>
> | Loss | Mask | Attn | C-Eval | MMLU-R | CMMLU | WG | HS | ARC-E | ARC-C | BBH | GSM8K | HEval | IFEval |
> |------|------|------|--------|--------|-------|------|------|-------|-------|------|-------|-------|--------|
> | *Teacher (1.7B)* |  |  | 62.8 | 66.9 | 62.0 | 54.2 | 59.8 | 84.8 | 73.6 | 34.3 | 80.9 | 58.5 | 66.1 |
> | SFT | C | frozen | 33.1 | 45.7 | 46.1 | 49.5 | 31.4 | 74.8 | 60.6 | 9.5 | 54.0 | 0.0 | 36.2 |
> | KD | C | frozen | 56.5 | **55.0** | 54.0 | 33.2 | 23.5 | 68.5 | 58.2 | 21.2 | **60.1** | 34.8 | 58.6 |
> | KD | P+C | frozen | 56.5 | 52.2 | 54.8 | **51.2** | **32.3** | **80.5** | **68.1** | 19.6 | 58.0 | 7.3 | 56.4 |
> | KD | C | unfrozen | **56.8** | 54.9 | **55.0** | 32.1 | 18.0 | 63.7 | 53.2 | **21.9** | 58.7 | **35.1** | **58.7** |
>
> _Perplexity-based (LM-eval):_
>
> | Loss | Mask | Attn | WG | HS | ARC-E | ARC-C | PIQA | LAMBADA (↓) | MMLU |
> |------|------|------|------|------|-------|-------|------|-------------|------|
> | *Teacher (1.7B)* |  |  | 61.6 | 60.5 | 69.7 | 42.8 | 72.6 | 12.2 | 55.4 |
> | SFT | C | frozen | **61.2** | **60.3** | 63.9 | 38.7 | 71.4 | **11.6** | 51.5 |
> | KD | C | frozen | 60.9 | 58.2 | 68.5 | 41.3 | 71.6 | 13.4 | 53.0 |
> | KD | P+C | frozen | 60.7 | 58.9 | 68.0 | 41.7 | **72.1** | 12.9 | **53.4** |
> | KD | C | unfrozen | **61.2** | 57.9 | **69.0** | **42.2** | 71.5 | 12.9 | 53.3 |
>
>
> We confirmed that two of the three conclusions reproduce cleanly. For the training objective, KD leads SFT on generation by 7.5pp on average, with the largest gaps on instruction following, code, and Chinese knowledge, while SFT attains the lower perplexity on the language-modeling benchmarks. This is the same direction of disagreement between the two protocols that we report at 0.6B. For attention freezing, the two KD, C rows of Table 1 (frozen versus unfrozen, otherwise identical) show freezing helping generation. The gains are large on the commonsense multiple-choice cluster (around 5pp each on HellaSwag and ARC), and on the remaining benchmarks the two runs sit within their seed-to-seed variation, so freezing never significantly underperforms. The average gain is 1.4pp across the eleven benchmarks. Perplexity moves by less than half a point and if anything slightly favours the unfrozen run, so here too the generation gain is largely invisible to perplexity. The loss-masking axis reproduces only in part, and we say so in the appendix. Completion-only keeps a large advantage on instruction following, code, and reasoning, but full-sequence masking overtakes it on the commonsense multiple-choice cluster (WinoGrande, HellaSwag, ARC), giving full-sequence a slightly higher generation average at 1.7B.

---

> > ### Author Response · Authors · 2026-05-27
> > **Response to reviewer mgKZ p2**
> >
> > ### Clarify the role of perplexity in the methodology (Important)
> >
> > We thank the reviewer for surfacing this tension, which is a fair one. The resolution is that perplexity plays two different roles in the paper. As a final-evaluation metric it judges a fully trained student on its completion tokens, asking how closely the student reproduces the teacher's generation, and this is the use where we document ranking inversions against generation-based evaluation. During layer selection it instead acts as a cheap comparative signal over candidate attention placements on an undertrained scaffold, asking only which layer positions matter for the architecture. The model being scored and the question being asked are different in the two cases, and we have added a two-sentence statement of this distinction to the layer-selection section.
> >
> > We agree that, with no compute constraint, a direct downstream-performance signal would be preferable even for layer selection. We did not use one because it is expensive, and in the appendix we now position our choice against two reference points already cited in the paper.
> >
> > - PostNAS (Gu et al., 2025) is the generation-aware upper bound, a once-for-all super-network scored with direct downstream-task signals (loss on the correct MMLU answer span, task accuracy on math and retrieval). Its attention-placement step alone consumes roughly 50B tokens, which is significantly above our whole distillation compute budget.
> > - GA-S2 (Li et al., 2025) scores each candidate layer by teacher-student KL after a per-variant distillation pass, totalling roughly 7 to 26B tokens at our scale.
> > - Our procedure removes per-candidate training entirely. We distil a 100M-token fully-linear scaffold once, then score beam-search candidates by held-out perplexity averaged over three probes, two of which are task-shaped (associative recall for retrieval, C-Eval for knowledge).
> >
> > Crucially, the cheap proxy does not cost us selection quality. At the five-attention-layer budget on Qwen3-0.6B (Appendix B), GA-S2 attains the lowest in-pipeline loss but not the strongest downstream performance, itself a within-pipeline echo of the perplexity-versus-generation disagreement we document at final evaluation, while our beam search matches or narrowly edges GA-S2 on aggregate downstream at roughly 100 times less search-stage compute. So a PostNAS-style downstream signal would be preferable given its compute, but in the budget regime our pipeline targets, beam search over probe-perplexity is the most efficient point on this spectrum that does not degrade selection quality.

---

> > > ### Author Response · Authors · 2026-05-27
> > > **Response to reviewer mgKZ p3**
> > >
> > > ### Expand architectural comparisons (Non-critical)
> > >
> > > We have broadened the architecture comparison. In addition to KDA and Mamba2, the revised comparison table and its perplexity-based companion now report three further sequence mixers distilled at 0.6B under the identical GenDistill pipeline: Gated DeltaNet (GDN), Gated Linear Attention (GLA), and Lightning Attention. These cover the main axes of the linear-attention design space, a delta-rule update in the same family as KDA (Gated DeltaNet), a gated-linear forget gate without a delta rule (GLA), and a data-independent slope-based decay (Lightning Attention). Every model shares the same layer selection, distillation stages, data, and hyperparameters, so the comparison isolates the sequence mixer. We note that the training hyperparameters were kept fixed and were not retuned for each mixer separately.
> > >
> > > **Table 3. Architecture comparison (generation-based, EvalScope), Qwen3-0.6B.** All students are hybrids retaining the 7 selected attention layers; we show only the hybrid variants here, as these are the configurations relevant to the comparison. Averaged over 3 seeds. Best student per column in **bold**. "--" = BBH not evaluated for Hybrid GLA and Hybrid Lightning.
> > >
> > > | Model | C-Eval | MMLU-R | CMMLU | WG | HS | ARC-E | ARC-C | BBH | GSM8K | HEval | IFEval |
> > > |-------|--------|--------|-------|------|------|-------|-------|------|-------|-------|--------|
> > > | *Teacher (Qwen3-0.6B)* | 42.9 | 46.1 | 45.2 | 51.2 | 39.0 | 70.8 | 55.0 | 27.4 | 57.5 | 35.3 | 57.1 |
> > > | Hybrid Mamba | 37.8 | 38.6 | 37.1 | 50.1 | 29.0 | 63.3 | 46.2 | 12.1 | 28.3 | 11.2 | 43.7 |
> > > | Hybrid KDA | 38.2 | **40.3** | **38.7** | **50.7** | **30.0** | **63.5** | **48.4** | **18.9** | **34.5** | **19.9** | **46.7** |
> > > | Hybrid GDN | 32.5 | 27.3 | 27.9 | 29.5 | 14.2 | 39.7 | 26.7 | 12.0 | 20.3 | 10.1 | 38.9 |
> > > | Hybrid GLA | **38.8** | 28.2 | 25.9 | 44.6 | 18.8 | 51.0 | 37.7 | -- | 19.0 | 1.5 | 33.6 |
> > > | Hybrid Lightning | 18.6 | 10.2 | 12.8 | 43.4 | 11.3 | 14.9 | 9.6 | -- | 9.6 | 0.0 | 15.7 |
> > >
> > > The broadened comparison reinforces our conclusions. KDA remains the strongest mixer, with its lead concentrated on the hardest reasoning and code-generation benchmarks. GLA and GDN trade off along the metric axis the paper emphasises. GLA is stronger on knowledge and common-sense tasks, while GDN is stronger on math and code generation. Both trail KDA and Hybrid Mamba on aggregate.
> > >
> > > Lightning Attention is an informative negative result as it also confirms the paper's central evaluation point. As Table 4 shows, under perplexity-based scoring its WinoGrande, HellaSwag, ARC, and PiQA accuracies sit within a few points of the other hybrids, while generation-based evaluation reveals a near-total collapse.
> > >
> > > **Table 4. Architecture comparison under perplexity-based evaluation (LM Evaluation Harness), Qwen3-0.6B.** Same models and seeds as Table 3. Accuracy (%) for all tasks except LAMBADA (perplexity, ↓). Best student per column in **bold**. "--" = CMMLU not evaluated for the teacher under LM-Eval.
> > >
> > > | Model | WG | HS | ARC-E | ARC-C | PIQA | LAMBADA (↓) | MMLU | CMMLU |
> > > |-------|------|------|-------|-------|------|-------------|------|-------|
> > > | *Teacher (Qwen3-0.6B)* | 55.5 | 47.3 | 55.6 | 34.2 | 65.9 | 24.7 | 40.2 | -- |
> > > | Hybrid Mamba | 54.0 | 42.9 | 53.6 | 30.5 | 65.7 | 73.0 | 34.7 | 38.8 |
> > > | Hybrid KDA | **57.2** | **45.2** | 52.5 | **31.7** | **66.3** | **38.0** | **40.6** | **40.0** |
> > > | Hybrid GDN | 53.3 | 41.6 | 51.2 | 28.7 | 64.0 | 56.1 | 30.0 | 34.9 |
> > > | Hybrid GLA | 54.1 | 43.6 | **54.3** | 30.5 | 65.7 | 51.9 | 38.0 | 39.2 |
> > > | Hybrid Lightning | 55.1 | 40.4 | 51.4 | 29.0 | 64.1 | 111.4 | 26.7 | 36.0 |

---

> > > > ### Author Response · Authors · 2026-05-27
> > > > **Response to reviewer mgKZ p4**
> > > >
> > > > ### Add qualitative analysis of generation behaviour (Non-critical)
> > > >
> > > > We agree the disagreement deserves a qualitative characterisation alongside the aggregate numbers, and we have added one in a new appendix section, "Qualitative KD vs SFT Failure Patterns,". We compared two same-recipe Stage-3b checkpoints (Hybrid-KDA 0.6B, completion-only, 500M tokens) that differ only in loss type, on two unrelated tasks (HellaSwag and GSM8K). Both show a large accuracy gap in KD's favour. We noticed four behaviours that occur in the SFT generations and are essentially absent from KD on the same prompts.
> > > >
> > > > 1. Spurious template wrapping. SFT wraps 100% of outputs in literal `<answer>...</answer>` tags on both tasks, including on GSM8K where the 4-shot prompt demonstrates a plain boxed format. KD wraps 0%. The tags appear in part of the upstream Stage-3b instruct corpus, and SFT has learned them as a literal output prefix and suffix, whereas KD trains against the teacher's full output distribution and the Qwen3 teacher does not emit the tag.
> > > > 2. Multiple-choice positional collapse. On HellaSwag, SFT picks option "A" on 79.5% of all questions against KD's 54%.
> > > > 3. Wordier and less accurate math chains. On GSM8K, SFT generates a mean of 404 output tokens against KD's 302. Looking at individual prompts and answers, the SFT chains suffer from restating the problem with extra ceremony, introducing ill-defined intermediate variables, and committing arithmetic errors that KD avoid on the same prompts.
> > > > 4. Few-shot recency mimicry. On the 4-shot GSM8K prompt, SFT regurgitates tokens lifted directly from the recency-most demonstration into answers to unrelated questions roughly three times as often as KD. Re-running with no in-context demonstrations drops the mimicry rate to zero on both models and shrinks the KD-versus-SFT gap by half, confirming that the demonstrations themselves are the trigger and that SFT is the more susceptible of the two.
> > > >
> > > > These four behaviours point to a single underlying mechanism. KD trains the student to match the teacher's full distribution over the next token, not just the single correct answer, so the student picks up soft behaviours that the hard label cannot convey, such as how the teacher spreads probability across the answer options and how little it leans on the in-context examples. SFT, which trains only against the single correct token, does not pass these on. Each of the individual patterns we observe also has an established precedent in the LLM literature, spanning miscalibrated multiple-choice selection (Zheng et al., 2024), label-copying from in-context demonstrations (Ali et al., 2024), sensitivity to few-shot prompt formatting (Sclar et al., 2023), verbatim emission of low-diversity training templates (Lin et al., 2025), and the flaw-repetition failure mode under SFT-only post-training of reasoning models (Yao et al., 2025). We include representative verbatim prompt-and-output traces for each of the four pathologies in the new appendix section, placing the SFT failure beside the KD output on the same prompt.
> > > >
> > > >
> > > > ### Further analyse long-context degradation (Non-critical)
> > > >
> > > > We agree the long-context degradation deserves a more substantive discussion.
> > > >
> > > > First, we would like to point out that the gap reflects the absence of a long-context training stage rather than an intrinsic limitation of the hybrid architecture. The Qwen3 teacher itself acquired its 32K capability through a dedicated final pre-training stage on long-context corpora (75% of that stage's data drawn from 16K to 32K sequences), further extended at inference with YARN and Dual Chunk Attention (Yang et al., 2025). Our students never see sequences beyond 4K, so a cliff past 16K is the predicted outcome of omitting that stage rather than evidence against the hybrid student.
> > > >
> > > > Second, concurrent work in cross-architecture distillation establishes that hybrid linear-attention students recover long-context quality once given such a targeted stage. HALO (Chen et al., 2026) matches or exceeds its Transformer teachers on needle-in-a-haystack retrieval at 128K after a dedicated 1B-token long-context finetuning stage, and MiniCPM-SALA (MiniCPM Team, 2026) scales the strategy to a 9B hybrid using a long-decay schedule that progressively extends context from 4K to 520K. Both works additionally adopt a small architectural modification, HyPE, which places the rotary position embedding on the recurrent and linear layers and removes it from the retained attention layers.
> > > >
> > > > HyPE is directly applicable to our hybrid student and complementary to the recipe choices we report. We did not repeat this modification and view it as a natural pairing with a long-context training stage in future work. The recipe contributions we report are orthogonal to both axes and apply unchanged.
> > > >
> > > > We have updated the limitations section and also updated the section on the long-context evals in the appendix to reflect this.

---

> > > > > ### Author Response · Authors · 2026-05-27
> > > > > **Response to reviewer mgKZ p5**
> > > > >
> > > > > ### Broader impact
> > > > >
> > > > > We thank the reviewer for the suggestion. We added a brief statement noting that more efficient distilled models lower the compute and memory cost of deployment, which can broaden accessibility and increase deployment scale, with the standard considerations that apply to large language models otherwise unchanged.

---

### Review · Reviewer_vr68 · 2026-05-14

**Summary Of Contributions:**

This paper presents a distillation pipeline for large language models, and an empirical study of various distillation choices, using perplexity and generation accuracy as metrics.

**Additional Comments:**

This is not my primary area of expertise, so I may have missed something important.

**Audience:**

Yes

**Audience Explanation:**

Yes- distillation is fundamentally a topic of great interest to the ML community, and understanding the limitations of different metrics is important.

**Claims And Evidence:**

Yes

**Claims Explanation:**

Yes, as far as I can tell the experiments make sense (basically a comparison of the same distillation procedure with different metrics), and the conclusions follow straightforwardly from the experiments.

**Requested Changes:**

My main requested change is about the narrative. Ultimately the paper seems to present two things which are fairly separate:

1. An empirical study of perplexity vs generation metrics, showing they don't agree.
2. Presenting a custom 3 stage distillation pipeline, GenDistill.

These are presented together in a very unclear way in my opinion- the beginning of the paper (and the title) emphasizes the metric part, then the first few sections mostly focus on the distillation pipeline, and the results section focuses on the metric. As far as I can tell the "GenDistill" pipeline doesn't actually use any generation-based metrics within it. Overall, the paper is a bit confusing to read end to end.

Also, the main results of the paper are presented as tables, where the trend of perplexity disagreeing with generation is not easy to see.

In its current form, the paper seems more suited to an LLM reader than a human reader. So, my request to the authors is to _either_:

A) make the paper more human readable: structure the contributions separately, make the interactions between the two parts of the paper clear, and turn some of the tables into figures where the trends can be seen visually.

B) lean into "this is a paper written for LLMs" and submit a markdown formatted paper which is more directly optimized for LLM consumption. TMLR has recently started allowing this.

---

> ### Author Response · Authors · 2026-05-27
> **Response to reviewer vr68**
>
> We thank the reviewer for an accurate diagnosis. The observation that the paper reads as two loosely connected pieces, a study of perplexity versus generation metrics and a custom distillation pipeline, was fair, and we have substantially revised the narrative to reconcile the two threads. We also converted the central results tables into figures, which are easier for a human reader to take in, and moved supporting material to the appendix. The revised manuscript foregrounds the empirical study and recasts the pipeline as the instrument that serves it, rather than a co-equal second contribution. The pipeline is the setup that lets us compare and ablate distillation design choices one at a time, and its end product is a concrete recipe for efficient distillation.
>
> The reviewer is also right that GenDistill uses no generation-based metric inside the training loop, and we have corrected our claim of generation-based evaluation used "throughout to guide every design decision," which overstated its role. Generation-based evaluation is the criterion by which we selected among the pipeline's design choices, so the resulting recipe is the product of those generation-guided decisions rather than of an in-loop training signal.
>
> We made the following concrete changes to the manuscript.
>
> - We reframed the abstract, introduction, and contribution bullets along this narrative.
> - We reorganized the results section to present the empirical findings first, followed by the architecture comparison (including the new 1.7B scale-up) and the efficiency experiments.
> - We moved several secondary results to the appendix and reference them from the main text (for example the Stage 3a budget table and its training-loss figure).
> - We replaced the central Stage 3b ablation table with a spider chart that shows the main differences visually. This omits the training-duration results, which we moved to the appendix as a separate figure.
> - We moved the background section introducing Kimi Delta Attention to the appendix.
>
> The reviewer also suggested making the paper more LLM-friendly by switching to a markdown format (Option B). We thank the reviewer for the suggestion, but have chosen Option A. Our aim is a paper a human reader can follow end to end, and the narrative restructure, the new figures, and the qualitative-failure appendix are all in service of that.

---

> > ### Comment · Reviewer_vr68 · 2026-06-04
> > **Great revisions!**
> >
> > The new paper is much clearer and I really liked figure 3. Thanks for the revisions!
> >
> > The suggestion to make the paper more LLMs was a slightly tongue-in-cheek suggestion: sorry if that wasn't clear (good-natured humour is hard to convey through text). I think the revisions to make it more human-readable was definitely the right option and I'm glad to see you put in the effort to do this. I have no remaining concerns.
> >
> > I did notice 1 typo: at the bottom of page 4, there seems to be a missing word at the start of the paragraph (maybe "The").

---

> > > ### Author Response · Authors · 2026-06-10
> > > **Fixed typo**
> > >
> > > We are glad to hear that the revisions improved the paper quality.
> > >
> > > Regarding the typo: it actually was not a typo, but the beginning "The" was stuck on a line that was before the figure. We fixed this and uploaded a new version of the paper.
> > >
> > > We also took the paper suggestion (i.e. making it LLM friendly) lightly. But you were correct to point out that there were many (large) tables that failed to convey the main message to the reader, so it was definitely valuable feedback.

---

### Review · Reviewer_ZGje · 2026-07-05

**Summary Of Contributions:**

This paper studies cross-architecture distillation from pretrained Transformers into efficient hybrid sequence models. Its main argument is that perplexity or likelihood-based evaluation can be misleading for distilled students, because a model may assign high likelihood to correct answers without being able to generate them autoregressively.

The authors propose GenDistill, a multi-stage pipeline for distilling Qwen3 teachers into Hybrid-KDA students. The pipeline is structurally close to prior staged distillation methods such as MOHAWK, but adapts the process to Kimi Delta Attention and uses generation-based evaluation to select recipe choices. The paper studies several axes: KD vs SFT, completion-only vs full-sequence masking, data choice, attention freezing, training duration, and sequence mixer choice.

Main strengths:
1. The PPL-vs-generation discrepancy is important and clearly motivated.
2. The ablations provide useful practical insights: KD, completion-only masking, teacher-aligned data, and freezing retained attention layers help generation quality.
3. The hybrid architecture comparison shows that retaining some attention layers is important for preserving reasoning/retrieval ability.

Main weaknesses:
1. The paper lacks a controlled MOHAWK-style baseline under the same teacher, student scale, token budget, and generation evaluation.
2. The novelty boundary relative to prior staged distillation work could be clearer.
3. Some presentation choices, especially Figure 3, make the key PPL-side comparisons hard to read.

**Additional Comments:**

Overall, this is a useful and timely paper. I view its main contribution as a generation-focused analysis and recipe for cross-architecture distillation, rather than a fundamentally new distillation framework.

**Audience:**

Yes

**Audience Explanation:**

The paper should interest readers working on efficient LMs, distillation, hybrid Transformer/SSM architectures, and evaluation. The main lesson is practically useful: distilled efficient models should be evaluated by actual generation, not only by likelihood ranking.

**Broader Impact Concerns:**

I do not see major broader impact concerns. The work may reduce inference cost and make model deployment more efficient. A short note on deployment-scale risks from cheaper generation would be sufficient.

**Claims And Evidence:**

Yes

**Claims Explanation:**

The main claim that perplexity-based evaluation can misrepresent generation quality is well supported. The QRWKV6 example is convincing, and the authors’ own ablations repeatedly show that recipe choices with similar likelihood scores can differ substantially under generation-based evaluation.

However, the evidence is stronger for the evaluation/recipe-selection claim than for any claim of method superiority over prior cross-architecture distillation pipelines. Since GenDistill follows a MOHAWK-style staged framework, a controlled baseline would be needed to isolate the contribution of the new recipe choices.

**Requested Changes:**

1. Clarify the novelty boundary relative to MOHAWK and related staged distillation methods. The paper acknowledges MOHAWK, but should more explicitly state what is inherited and what is new: Hybrid-KDA adaptation, instruction-stage KD, and generation-based recipe selection.

2. Add a controlled prior-method baseline if feasible. The most useful comparison would be a MOHAWK-style pipeline under the same teacher, student scale, token budget, and generation-based evaluation. If this is too expensive, the paper should explicitly avoid implying superiority over prior staged distillation methods.

3. Improve the presentation of Figure 3. The PPL-side curves overlap heavily, making it hard to see differences or ranking changes. A compact table with per-benchmark scores, average deltas, and ranking reversals would better support the central argument.

4. Clarify the distinction between long-context efficiency and long-context quality. The hybrid model improves memory and runtime behavior, but its long-context task performance still trails the teacher without dedicated long-context training.

---

> ### Author Response · Authors · 2026-07-14
> **Response to reviewer ZGje**
>
> We thank the reviewer for the careful and constructive review, and we have revised the
> paper in response. We respond to each point below.
>
> ### Clarify the novelty boundary relative to MOHAWK
>
> GenDistill is similar in spirit to MOHAWK's pipeline and inherits its bottom-up, multi-stage structure: progressively align the student to the teacher, then perform end-to-end knowledge distillation. It also differs from MOHAWK in several practical respects, including the choice of student sequence mixer, the Stage 1 loss (forced by the student mixer, since KDA cannot materialize the attention matrix that MOHAWK aligns), the per-stage token budgets, and the dataset choices. Our contribution is the empirical analysis of these design choices under generation-based evaluation and the resulting recipe, and each choice we make is grounded in a controlled experiment showing improved downstream generation. We have revised the manuscript to keep this scope explicit: in the Limitations section we mention the limitation of comparing full pipelines head to head, and that we limit our claims to the generation-based evaluation analysis and the resulting recipe rather than making concrete claims on pipeline superiority.
>
> To clarify the boundary relative to other staged distillation pipelines beyond MOHAWK, we have added a compact comparison table to the Related Work section (Table 1 in the revised manuscript) that contrasts MOHAWK, Mamba-in-Llama, Llamba, and RADLADS with our pipeline across the axes that separate these methods: the student sequence mixer, hybrid retention versus full conversion, the stage sequence, the token budget, the form of any instruction stage, and the headline evaluation protocol. The table makes explicit what we inherit, the bottom-up align-then-distill blueprint common to this line of work, and what is specific to GenDistill: the hybrid Kimi Delta Attention student, the completion-only-masked instruction-distillation stage on teacher-aligned data, and the generation-based evaluation.
>
> ### Add a controlled prior-method baseline, or avoid implying superiority
>
> We have taken the second option and revised the manuscript to avoid implying pipeline-level superiority. The reason connects to the point above: a distillation pipeline has many entangled degrees of freedom (the teacher and its architecture, the student sequence mixer, the per-stage losses and their token budgets, the retained attention layers, and the instruction dataset, which itself depends on the teacher model), so a head-to-head comparison of whole pipelines cannot attribute any quality difference to a single cause. This entanglement is actually what motivated this paper in the first place, to compare the individual choices inside a distillation pipeline in a more controlled way.
>
> We also want to point out that the requested controlled MOHAWK-style baseline is very similar to our Hybrid Mamba and Pure Mamba rows in Table 4 of the manuscript. The comparison there is at the same token budget, same teacher, same student scale (Pure Mamba baseline uses the same sequence mixer as in the MOHAWK paper), using the same datasets. The only way our setup departs from a literal MOHAWK controlled baseline is the Stage 1 loss: MOHAWK aligns the full attention matrix, whereas we align hidden states. Our table shows that the pipeline with KDA outperforms the Mamba variants. For precision, we also note that the table labels these rows "Mamba", but the model is Mamba2 rather than Mamba1, and we have added a footnote to the table clarifying this. Nevertheless, this architecture ablation experiment cannot be used to imply superiority between the two pipelines, since these prior pipelines were tuned for substantially larger token budgets than ours (e.g. Llamba uses around 8B tokens for distillation, while we use 1B). The simple reason for not running the longer baseline is our compute budget, with the deeper reason being that this experiment is not really needed to convey our main conclusions.

---

> > ### Author Response · Authors · 2026-07-14
> > **Response to reviewer ZGje**
> >
> > ### Improve the presentation of Figure 3
> >
> > We agree the perplexity-side polygons overlap heavily. Rather than remove the figure (a prior reviewer specifically asked us to turn this content into a figure so the trend is visible), we have added a compact companion table beside the spider plot, so Figure 3 now has two panels: the spider on the left and a small table on the right. The table reports, for the four configurations and the teacher, the average accuracy over the six benchmarks common to both protocols, under perplexity and under generation, with the gap to the teacher shown in red. The full per-benchmark numbers remain in the appendix, and the paper points to that appendix table both in the sentence introducing the figure and in the figure caption. The per-benchmark ranking reversals are visible in that appendix table: on HellaSwag, ARC-C, and MMLU the SFT configurations rank at or near the top under perplexity yet fall to the bottom under generation, whereas on the remaining shared benchmarks the four configurations are nearly tied under perplexity.
> >
> > ### Clarify long-context efficiency versus long-context quality
> >
> > We have made this distinction explicit and concise in the Discussion. The hybrid wins decisively on long-context efficiency (memory, time-to-first-token, and decode throughput), whereas its long-context quality (task accuracy and retrieval) still trails the teacher in the absence of a dedicated long-context training stage. This quality gap reflects the missing long-context stage rather than an architectural limit, since the teacher itself acquired its 32K capability through a dedicated long-context phase, and concurrent work recovers long-context quality once such a stage is added.
> >
> > ### Broader impact
> >
> > We have added a note to the Broader Impact Statement that cheaper and faster generation also lowers the cost of producing text at scale, so the standard misuse considerations for large language models, such as large-scale generation of low-quality or misleading content, apply here as well, with efficiency changing the scale of these risks rather than their qualitative nature.